# Structural and functional insights into the interaction between Ku70/80 and Pol X family polymerases in NHEJ

Philippe Frit [1,5], Himani Amin[2,5], Sayma Zahid [2,5], Nadia Barboule[1,5], Chloe Hall[2], Gurdip Matharu [2], Steven W. Hardwick [3], Jeanne Chauvat [1], Sébastien Britton [1], Dima Y. Chirgadze [3], Virginie Ropars [4], Jean-Baptiste Charbonnier [4], Patrick Calsou [1]✉ & Amanda K. Chaplin [2]✉

Non-homologous end joining (NHEJ) is the main repair pathway for double-strand DNA breaks (DSBs) in mammals. DNA polymerases lambda (Pol λ) and mu (Pol μ), members of the Pol X family, play a key role in this process. However, their interaction within the NHEJ complexes is unclear. Here, we present cryo-EM structures of Pol λ in complex with the DNA-PK long-range synaptic complex, and Pol μ bound to Ku70/80-DNA. These structures identify interaction sites between Ku70/80 and Pol X BRCT domains. Using mutants at the proteins interface in functional assays including cell transfection with an original gap-filling reporter, we define the role of the BRCT domain in the recruitment and activity of the two Pol X members in NHEJ and in their contribution to cell survival following DSBs. Finally, we propose a unified model for the interaction of all Pol X members with Ku70/80.

Non-homologous end joining (NHEJ) is the predominant pathway in which double strand DNA (dsDNA) breaks are repaired in mammals. Central to the process of NHEJ are large, multi-protein complexes formed by the canonical proteins DNA-PKcs, the heterodimer of Ku70/80, DNA ligase IV (LigIV), X-ray repair cross-complementing protein 4 (XRCC4) and XRCC4-like factor (XLF)[1]. In addition, PAXX (Paralog of XRCC4 and XLF) is an accessory NHEJ protein identified more recently which has functional redundancy with XLF in the NHEJ mechanism[2,3]. Although this repair pathway is a complex process, it is generally considered to proceed via three major steps; DNA end recognition, DNA end processing and finally DNA ligation[4].

The initial step of dsDNA break recognition relies predominantly on the Ku70/80 heterodimer, which engages the DNA ends and subsequently recruits the DNA-PKcs kinase, together forming the DNA-PK holoenzyme[5]. Recent cryo-electron microscopy (cryo-EM) structures of DNA-PK assemblies revealed how DNA substrates can be positioned for efficient synapsis via dimerization of this large enzyme. Intriguingly, DNA-PK has two distinct mechanisms to form a pre-synaptic complex, corresponding to alternate structural dimers of DNA-PK. It was first proposed that DNA-PK can synapse the broken DNA ends using a dimer of DNA-PK mediated by the C-terminal region of Ku80 (herein termed Ku80-mediated dimer)[6]. Soon after, an alternative DNA-PK dimer complex was described, which can form upon addition of XRCC4, LigIV and XLF[2,7,8]. In this second complex, the DNA-PK dimer is crucially bridged by XLF (herein termed XLF-mediated dimer). In both assemblies, the distance between the damaged DNA ends is identical at 115 Å. This distance is not close enough for DNA ligation to proceed without further rearrangement of the assembly, therefore both dimeric assemblies have been referred to as long-range synaptic complexes (LRC). We have shown recently that PAXX can replace XLF for bridging the Ku80-mediated DNA-PK dimer through binding to Ku70 but does not replace XLF in the XLF mediated dimer[2].

[1]Institut de Pharmacologie et Biologie Structurale (IPBS), Université de Toulouse, CNRS, Université Toulouse III—Paul Sabatier (UT3), Toulouse, France. [2]Leicester Institute for Structural and Chemical Biology, Department of Molecular and Cell Biology, University of Leicester, Leicester, UK. [3]Department of Biochemistry, Sanger Building, University of Cambridge, Cambridge, UK. [4]Institute for Integrative Biology of the Cell (I2BC), Institute Joliot, CEA, CNRS, Université Paris-Saclay, Gif-sur-Yvette, France. [5]These authors contributed equally: Philippe Frit, Himani Amin, Sayma Zahid, Nadia Barboule. ✉e-mail: calsou@ipbs.fr; ac853@leicester.ac.uk

The second step in the NHEJ mechanism is end processing and is crucial if the DNA ends cannot be readily ligated. Different end processing factors can be recruited to either trim or extend the damaged ends via the action of DNA nucleases or polymerases, as detailed below.

The final step of the repair mechanism involves ligation of the phosphodiester backbone with 5'-phosphate and 3'-hydroxyl moieties by LigIV, facilitated by XLF and XRCC4. To enable ligation, the LRC is thought to transition to a short-range complex (SRC) following autophosphorylation and removal of DNA-PKcs[9]. In this SRC, the damaged DNA ends are in much closer proximity to each other and thereby are primed for DNA ligation. A cryo-EM structure of the SRC has been solved which shows the catalytic domain of LigIV interacting directly with the DNA[7]. While structural studies have significantly improved understanding of the NHEJ mechanism by illuminating the architecture of several complexes central to the process, how the end processing proteins, in particular polymerases, interact with these complexes has remained an enigma.

The end processing factors known to be implicated in NHEJ include nucleases such as Artemis which can trim DNA nucleotides, and polymerases which can be used for end filling or extension. Recent structural data has revealed how Artemis can engage with a DNA-PK monomer via its N-terminal nuclease domain and is positioned between the N-HEAT (Huntingtin, Elongation Factor 3, A subunit of protein phosphatase 2 A, Target of Rapamycin/TOR) and M-HEAT domain of DNA-PKcs[10]. To date however, the structural mechanism of interaction between DNA polymerases and the NHEJ machinery has not been defined. The polymerases involved in NHEJ belong to the DNA Polymerases X family and catalyze the addition of nucleotides at the 3'-OH end of DNA. In mammals, three polymerases, namely Pol λ, Pol μ and terminal deoxynucleotidyl transferase (TdT), account for the majority of DNA synthesis during NHEJ, with the latter mainly involved in V(D)J recombination[11–13]. These polymerases share a common domain organization which consists of a breast cancer gene 1 (BRCA1) C-terminal (BRCT) region at the N-terminus followed by a C-terminal catalytic domain (Supplementary Fig. 1a). The catalytic region facilitates the protein:DNA interaction, thus enabling the function of the enzyme. The chemical nature of the DNA ends is a key determinant of the polymerase activity, with each of the three enzymes having varying degrees of template-dependency. Pol λ appears to display strong activity on DNA ends that have a paired primer terminus[14]. Furthermore, due to its accuracy, it has been proposed that Pol λ is given priority for gap filling in most cell types[15]. The BRCT domains are thought to direct the polymerases to sites of DNA damage via interactions with complexes comprising DNA and Ku70/80, or larger assemblies including the NHEJ proteins DNA-PKcs, LigIV and XRCC4[16,17]. While the structure of the BRCT and catalytic domain of these polymerases have been solved in isolation[16,18], how these domains engage with each other, and larger NHEJ complexes has remained unclear.

In order to elucidate how the polymerases interact with NHEJ machineries, we collected cryo-EM data of the Ku80 mediated dimer of DNA-PK in complex with Pol λ, and Ku70/80-DNA with Pol μ. From our structural data we identify the specific binding region of the BRCT domain of Pol λ and μ with the bridge region of Ku70/80. Using mutagenesis studies in cells on both Polymerases and Ku70/80, we identified key residues that are critical for Pol λ and μ anchoring in NHEJ complexes, efficient gap-filling activity and ultimately cell survival to DSBs. Finally, we propose that all members of the Polymerase X family share a conserved recognition site on the Ku70/80 heterodimer.

## Results

### Cryo-EM structure of Pol λ bound to the Ku80-mediated DNA-PK dimer

In order to understand how the polymerases λ and μ interact with the NHEJ LRC, proteins were first expressed and purified, as shown in Supplementary Fig. 1b. Electromobility shift assays (EMSAs) were first carried out with Ku70/80 and either Pol λ or Pol μ. As previously shown and predicted the DNA band could be seen to shift first upon binding to Ku70/80 and then shifted further once either polymerase concentration was increased. This first confirmed the ability of our purified polymerases to interact with Ku70/80 (Supplementary Fig. 1c, d).

We then attempted to study Pol λ, to initially observe binding to Ku70/80. We first used a simplified system of only Ku70/80, DNA, and Pol λ and collected cryo-EM data. Within this dataset we were only able to resolve Ku70/80 and no additional density for Pol λ. We therefore wondered whether it required the stability of other additional NHEJ factors. Thus, we then prepared and collected data of a sample containing DNA-PKcs, Ku70/80, DNA, XRCC4, LigIV, PAXX, and Pol λ to allow formation of the Ku80-mediated dimer of DNA-PK. Following extensive particle classification, a consensus map was obtained of the Ku80-mediated dimer of DNA-PK. The particles comprising this initial map were further characterized to generate maps of the Ku80-mediated dimer with and without XRCC4 and DNA LigIV (thereafter termed LX4) engaged (Fig. 1, and Supplementary Figs. 2–4). At this stage, it was apparent that there were no large areas of additional density that could accommodate the full-length Pol λ protein, when compared to previous maps of this DNA-PK dimer. However, a small region of additional density was apparent at the bridge that defines the thinner part of the Ku70/80 ring region (Supplementary Fig. 5). To focus specifically on this region, particles were re-extracted in smaller boxes corresponding to the monomeric size of DNA-PK, and maps were generated of single protomers from within the dimeric assemblies (herein termed half-dimers) with and without LX4 bound.

The structure without LX4 bound has an overall resolution of 4.53 Å and the map with LX4 bound has an overall resolution of 4.26 Å (Supplementary Figs. 2–5). Within both of these half-dimer maps, DNA-PK along with the PAXX Ku-binding motif (P-KBM) could be docked (Fig. 1). The PAXX P-KBM can be seen interacting with the von Willebrand-like (vWA) domain of Ku70, as has been previously characterized[2]. Additionally, XRCC4 and BRCT tandem repeats of LigIV can be docked into the map with LX4 bound (Fig. 1). As seen in our previous structures there is a central helix in DNA-PKcs which blocks the DNA ends, which is only present when LX4 is engaged. Additional density can be observed at the bridge formed by Ku70 and Ku80 in both maps, into which the NMR structure of the Pol λ BRCT domain (PDB: 2JW5) could be confidently docked (Fig. 1 and Supplementary Figs. 2–5). Although the full-length Pol λ was used, only the BRCT domain could be modeled into the cryo-EM map. This suggests that the catalytic domain has no stable interaction with DNA with the LRC in the cryo-EM assemblies determined.

### Molecular basis of the Pol λ BRCT interaction with Ku70/80

The Pol λ BRCT domain is positioned at the interface formed by Ku70 and Ku80, at the periphery of the DNA binding channel formed by the Ku70/80 heterodimer. An interaction between Pol λ and Ku70/80-LigIV-XRCC4 has been previously suggested, with the residues proposed to be involved in the interaction being situated in α1 helix of the BRCT domain[16]. In agreement with this previous study, our structure shows that Pol λ BRCT domain is positioned to allow helix α1 to dock into a groove formed between Ku70 and Ku80 (Fig. 2d). Specifically, the amino acid residues Arg57 and Leu60 of Pol λ BRCT domain appear to mediate the contact with Ku70/80, in agreement with previous data, which found that mutations of these residues prevented Pol λ interaction on DNA with Ku70/80 or Ku70/80-LigIV-XRCC4[16,17]. With regards to Ku70/80, the specific interaction sites encompass residues between 290–310 on Ku80 and residues 301–308 on Ku70.

### Characterization of key residues at Pol λ:Ku interface for Pol λ BRCT live recruitment to nuclear laser damage sites

Based on our structural data, we probed how the Pol λ BRCT interaction with Ku70/80 impacts the response to DNA damage in cells. We

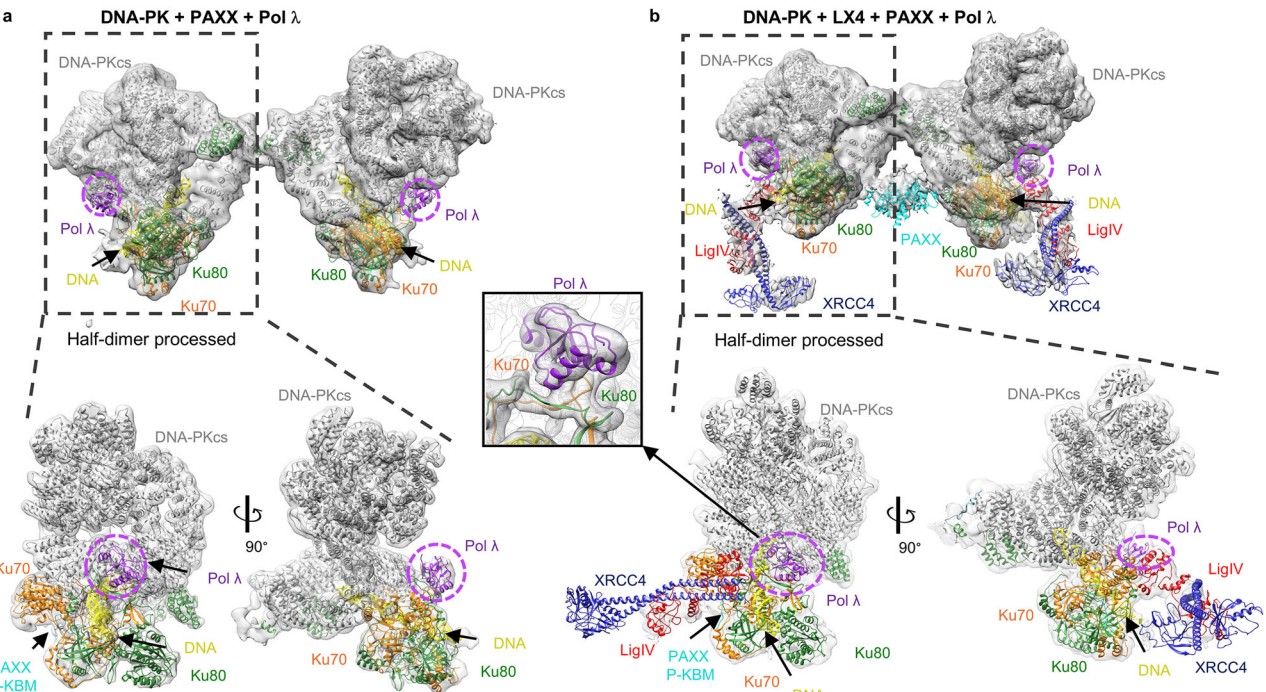

**Fig. 1 | Cryo-EM structures of DNA-PK, PAXX, and Pol λ with and without LX4.**
**a** DNA-PK + PAXX + Pol λ dimeric map with extra density for Pol λ highlighted in a dashed purple circle, with two orientations of processed half dimer presented below. DNA-PKcs in gray, Ku70 in orange, Ku80 in green, DNA in yellow, PAXX in cyan, and Pol λ in purple. **b** DNA-PK + PAXX + Pol λ + LX4 with extra density for Pol λ highlighted in a dashed purple circle, with two orientations of processed half dimer presented below. DNA-PKcs in gray, Ku70 in orange, Ku80 in green, DNA in yellow, LigIV in red, XRCC4 in dark blue, PAXX in cyan, and Pol λ in purple. Inset, zoom in of the Pol λ extra density. Source data are provided as a Source Data file.

first analyzed the influence on recruitment of Pol λ at laser-induced DNA damage sites. To establish to what extent Pol λ mobilization to damaged sites was Ku-dependent, we used in-house engineered U2OS cells that circumvent the lethality of Ku loss in human cells as previously described[2,19] (Fig. 2 and Supplementary Fig. 6a). Briefly, endogenous Ku70 expression was first knocked-down via the constitutive expression of an shRNA associated with cell rescue through expression of Ku70 tagged with a mini-auxin inducible degron (mAID). Upon addition of auxin (indole-3-acetic acid, IAA), degradation of endogenous Ku70 occurs within a few hours, with concomitant Ku80 depletion due to the known reciprocal stabilization of both Ku subunits. Full-length Pol λ fused to GFP was rapidly and substantially recruited to DNA damage sites in the presence of Ku. This recruitment of the full-length polymerase was only partly Ku-dependent since it substantially persisted under Ku depletion conditions (Fig. 2b, −red line). This may indicate an ability of Pol λ to interact directly with DNA at sites of damage, or may be related to Ku-independent repair functions of Pol λ outside NHEJ[20].

In contrast to the data obtained with full-length Pol λ, when the fusion was restricted to the N-terminal Pol λ BCRT domain (Supplementary Fig. 1a), its recruitment to micro-irradiated areas was mostly Ku-dependent since it was nearly abolished without Ku (Fig. 2c). Guided by our structural data, we next probed the impact of specific Pol λ BRCT mutations on the recruitment to DNA damage sites (Fig. 2d). As shown in Fig. 2e, all the mutants tested impaired GFP-Pol λ BRCT accrual at laser-induced DNA damage sites to various extents, with mutations at R57 or L60 positions being the most detrimental.

Notably, while GFP-Pol λ expression was nuclear with a nucleolar enrichment, the latter disappeared upon Ku depletion (Supplementary Fig. 6b). The Pol λ BRCT nucleolar enrichment was also strongly reduced with all BRCT mutants tested (Supplementary Fig. 6c). Since it is known that nuclear Ku is enriched in the nucleolus in the absence of DNA damage[21], this suggests that GFP-Pol λ localization to the

nucleolus relies on its interaction with Ku, and that this localization is disrupted by specific mutations within the BRCT domain of Pol λ.

To identify key residues of Ku70/80 at the Pol λ BRCT:Ku interface, we replaced endogenous Ku subunits with mutated forms of either Ku70 or Ku80 as guided by our structural data (Fig. 3a and Supplementary Fig. 7a–c). We then monitored the impact on Pol λ BRCT recruitment following DNA damage. To ensure that mutants of Ku70/80 did not influence Ku mobilization at damaged sites per se, we monitored simultaneously the recruitment of mCherry-PAXX co-expressed within the same cells, since PAXX recruitment is strictly Ku-dependent[2] (Fig. 3b–f). Mutations on the positions F303, T307, L310 on Ku70 or E292, E304 on Ku80 nearly abolished Pol λ recruitment (Fig. 3d, f). Notably, some substitutions on these positions (e.g., Ku70 L310R, T307A, Ku80 E304A) also affected PAXX recruitment, although at an intermediate extent compared to the full defect observed for Pol λ BRCT (Fig. 3c–e). Considering that PAXX binding to Ku70 is far away from the Ku bridge region, this indicates that these substitutions likely compromise the stability of the Ku:DNA interaction in addition to directly affecting the Ku:polymerase interaction. Nevertheless, these data establish that Ku70 F303, T307, L310, and Ku80 E292, E304 are key positions for Ku interaction with Pol λ.

### Set up of a gap-filling assay in cells

Pol λ has a large spectrum of substrates, accommodating breaks with overhangs (<4 to 12 nt), microhomologies (1–6 nt), and gaps (1–8 nt)[22]. Based on these features and to assess the impact of mutations at the Ku:Pol λ interface on Pol λ activity during end-joining, we designed a reporter plasmid, which once transfected in HEK-293T cells can be used as a gap-filling dependent DNA repair reporter (Fig. 4a). Briefly, the reporter is designed such that the Cas family member Cpf1 (Cas12a) induces two staggered DSBs in the mCherry cDNA which have two complementary nucleotides at the very end. Gap-filling at the junction restores mCherry expression, while end-filling enables the expression

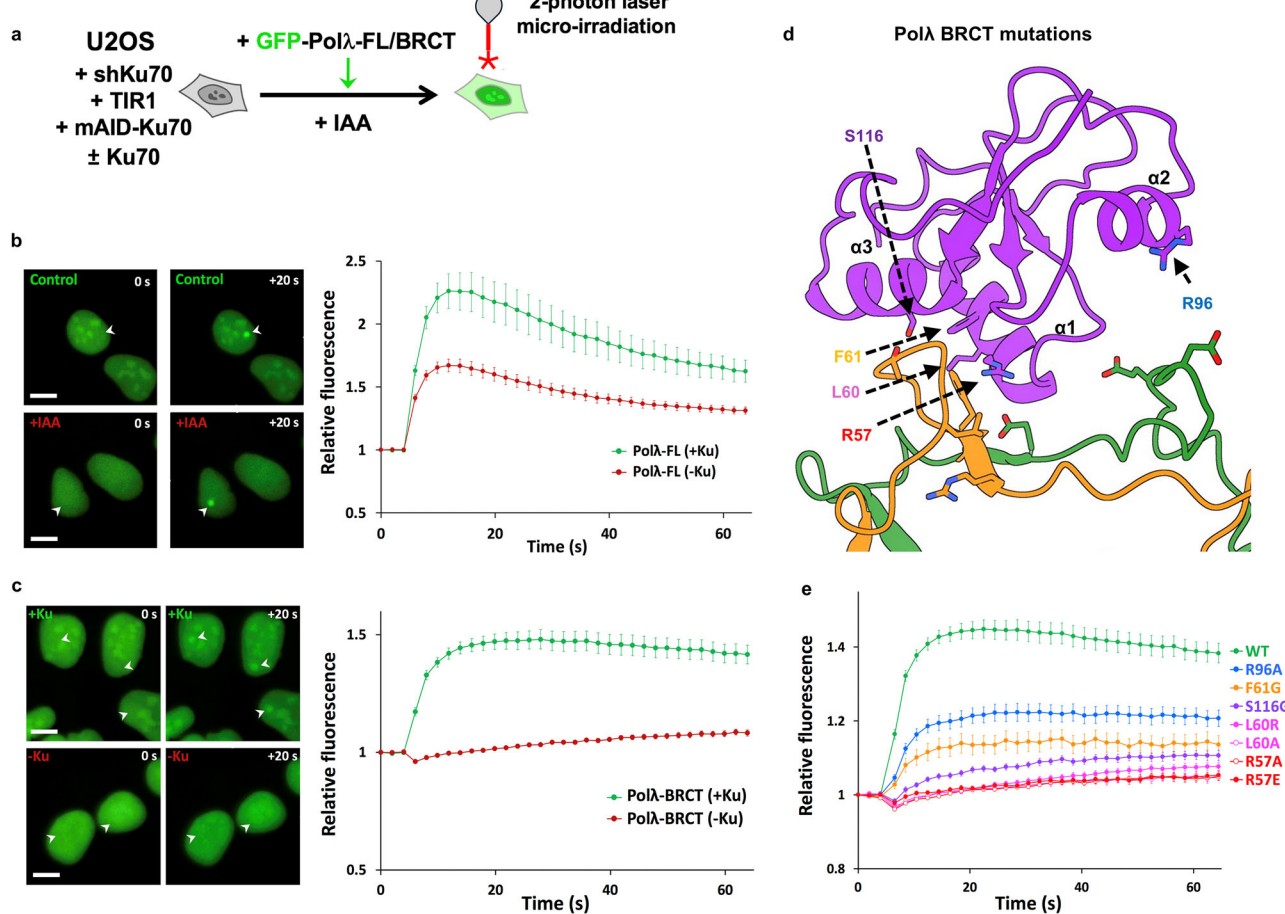

**Fig. 2 | Impact of mutations in the BRCT domain of Pol λ on its recruitment to DSBs by Ku. a** Principle of the laser micro-irradiation experiment. U2OS cells engineered for auxin (IAA)-induced Ku70 knockdown, rescued or not with wild-type (WT) Ku70 and expressing either GFP-tagged full-length Pol λ or its BRCT domain, were micro-irradiated with an 800 nm multiphoton laser to generate DSBs in subnuclear areas. **b** Left panel: representative images before and 20 s after irradiation (irradiated areas are indicated by white arrows) of nuclei from U2OS expressing GFP-tagged full-length Pol λ and depleted or not of Ku70 (±Ku). Scale bars represent 10 μm. Right panel: quantification of fluorescence accumulation at laser-induced DNA damage sites. Results are plotted as mean values ± SEM of n = 21 and n = 34 irradiated nuclei in the presence or absence of Ku, respectively. **c** Same as in (**b**) with U2OS expressing GFP-tagged Pol λ-BRCT. Results are plotted as mean values ± SEM of n = 27 and n = 22 irradiated nuclei in the presence or absence of Ku, respectively. **d** Position of mutated residues in the BRCT domain of Pol λ (purple) at the interface with Ku70/Ku80 (orange and green, respectively). **e** Quantification of fluorescence accumulation at laser-induced DNA damage sites in U2OS cells expressing GFP-tagged WT or mutated BRCT domain of Pol λ. Results are plotted as mean values ± SEM of 20 to 47 irradiated nuclei (Polλ-BRCT-WT: n = 47; -R57A: n = 20; -R57E: n = 20; -L60A: n = 23; -L60R: n = 21; -F61G: n = 20; -R96A: n = 22; -S116G: n = 22). Source data are provided as a Source Data file.

of a downstream EGFP cDNA which lies in a different reading frame. No expression of mCherry or GFP is expected if ends are trimmed or not ligated. Expression of BFP from a co-transfected plasmid accounts for transfection efficiency. Following Cpf1-induced DSB, expression of fluorescent proteins is measured by flow cytometry (Fig. 4b). We first established that gap-filling dependent mCherry expression relied on cells being transfected by the complete Cpf1/gRNA system (Fig. 4c), and on NHEJ activity since it was largely inhibited by an inhibitor of DNA-PK catalytic activity (NU7441), or upon Ku removal or genetic ablation of the NHEJ factors DNA-PKcs, LigIV, XLF or PAXX involved in break ends tethering and end joining[2] (Supplementary Fig. 7d, e). Moreover, sequencing the junctions from mCherry positive cells revealed that the expected product from gap-filling repair was formed, further validating our reporter system (Supplementary Fig. 7f). Notably, no GFP expression was detected implying that gap-filling is far dominant over end-filling at these staggered DSB (Fig. 4c). Then, we evaluated the Pol λ-dependency of the gap-filling activity detected. Pol λ was found dominant over Pol μ in gap-filling at three-nucleotide gaps on 5′ overhang substrates[14,23]. Indeed, Pol λ knock-out (KO) led to a ~70% reduction of the activity on our substrate that was fully restored

upon re-expression of WT Pol λ, while Pol μ KO alone did not impair gap-filling activity (Supplementary Fig. 7g, h). Notably, Pol μ KO in Pol λ KO cells further decreased gap-filling activity, suggesting a small compensation by Pol μ in the absence of Pol λ. Finally, expression of a catalytically dead Pol λ (D427A-D429A[24]) (Pol λ dead) further decreased the remaining gap-filling activity in Pol λ KO cells, supporting a dominant negative effect of the Pol λ dead construct on backup gap-filling enzymes, including Pol μ.

## Pol λ-dependent gap-filling activity relies on key positions in the Pol λ BRCT:Ku interface

Using the mainly NHEJ- and Pol λ-dependent gap-filling assay described above, we then analyzed the impact of mutations in Pol λ BRCT though transfection of Pol λ KO HEK-293T cells complemented with WT or mutant Pol λ. As a control, the same cells were used to evaluate direct end-joining (EJ) activity at blunt-ended breaks using our dedicated Cas9-based reporter plasmid assay described previously[19]. As shown in Fig. 5a, except for R96A, all other mutations tested in Pol λ BRCT reduced gap-filling efficiency at staggered DSB. The effect of F61G mutation is inconclusive since it lowered the protein expression

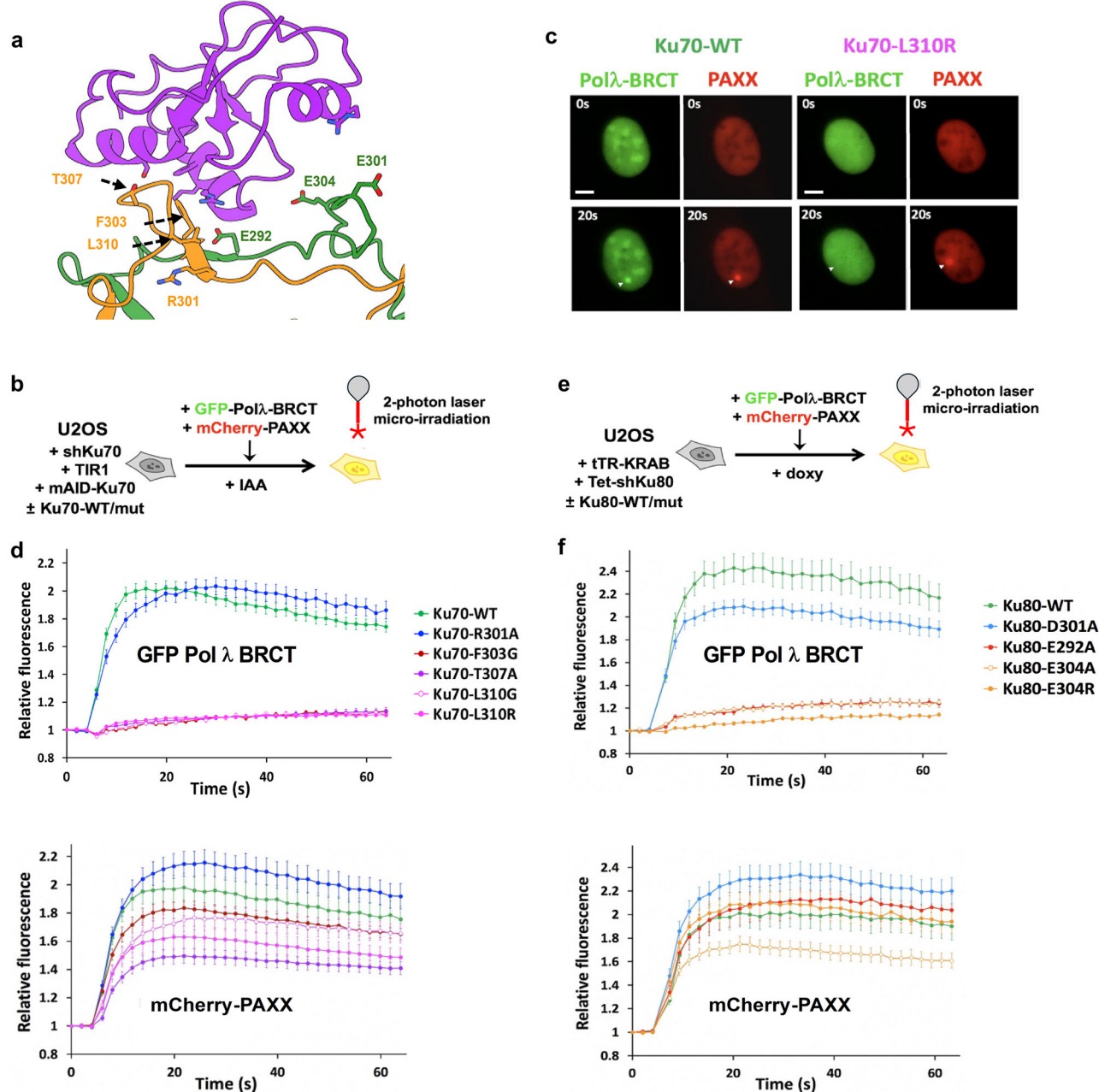

**Fig. 3 | Impact of mutations in the bridge region of Ku70 and Ku80 on the recruitment of the BRCT domain of Pol λ at DSBs. a** Position of mutated residues in the bridge domain of Ku70 (orange) or Ku80 (green) at the interface with Pol λ (purple). **b** Principle of the laser micro-irradiation experiment. U2OS cells engineered for auxin (IAA)-induced Ku70 knockdown, rescued with WT or mutated forms of Ku70 and expressing both the GFP-tagged BRCT domain of Pol λ and the mCherry-tagged PAXX protein, were micro-irradiated and accumulation of each fluorescence was analyzed as in Fig. 2. **c** Representative images before (upper frames) and 20 s after irradiation (lower frames, a white arrow indicates irradiated area) of nuclei from U2OS expressing either WT Ku70 (left) or the L310R mutant (right). Scale bars represent 10 μm. **d** Quantification of fluorescence accumulation at laser-induced DNA damage sites of GFP-Polλ-BRCT (upper chart) and mCherry-PAXX (lower chart) in U2OS cells expressing the indicated Ku70 constructs. Results

are plotted as mean values ± SEM of 20 to 38 irradiated nuclei (Ku70-WT: n = 24; -R301A: n = 32; -F303G: n = 38; -T307A: n = 21; -L310A: n = 21; -L310R: n = 20). **e** Principle of the laser micro-irradiation experiment. U2OS cells engineered for doxycycline (doxy)-induced Ku80 knockdown, rescued with WT or mutants of Ku80 and expressing both the GFP-tagged BRCT domain of Pol λ and the mCherry-tagged PAXX protein, were micro-irradiated and accumulation of each fluorescence was analyzed as in (**d**). **f** Quantification of fluorescence accumulation at laser-induced DNA damage sites of GFP-Polλ-BRCT (upper chart) and mCherry-PAXX (lower chart) in U2OS cells expressing the indicated Ku80 constructs. Results are plotted as mean values ± SEM of 20–27 irradiated nuclei (Ku80-WT: n = 20; -E292A: n = 25; -D301A: n = 27; -E304A: n = 27; -E30AR: n = 27). Source data are provided as a Source Data file.

(Fig. 5b). Notably, the extent of defect observed in gap-filling activity of full-length mutant Pol λ correlates with that observed in recruitment of the corresponding mutant GFP-BRCT at laser sites (Fig. 2e), again with R57 and L60 positions being the most crucial. Since no detectable repair defect was found at blunt-ended DSB, a readout for direct EJ, this

indicates that outside its catalytic function at defined breaks Pol λ is unlikely to fulfill a general function in the overall NHEJ complex assembly and/or stability. Using the same assays, we then evaluated the impact of mutations of Ku70 in the Pol λ:Ku interface by using HEK-293T cells expressing mAID-Ku70 fusion and transduced with WT or

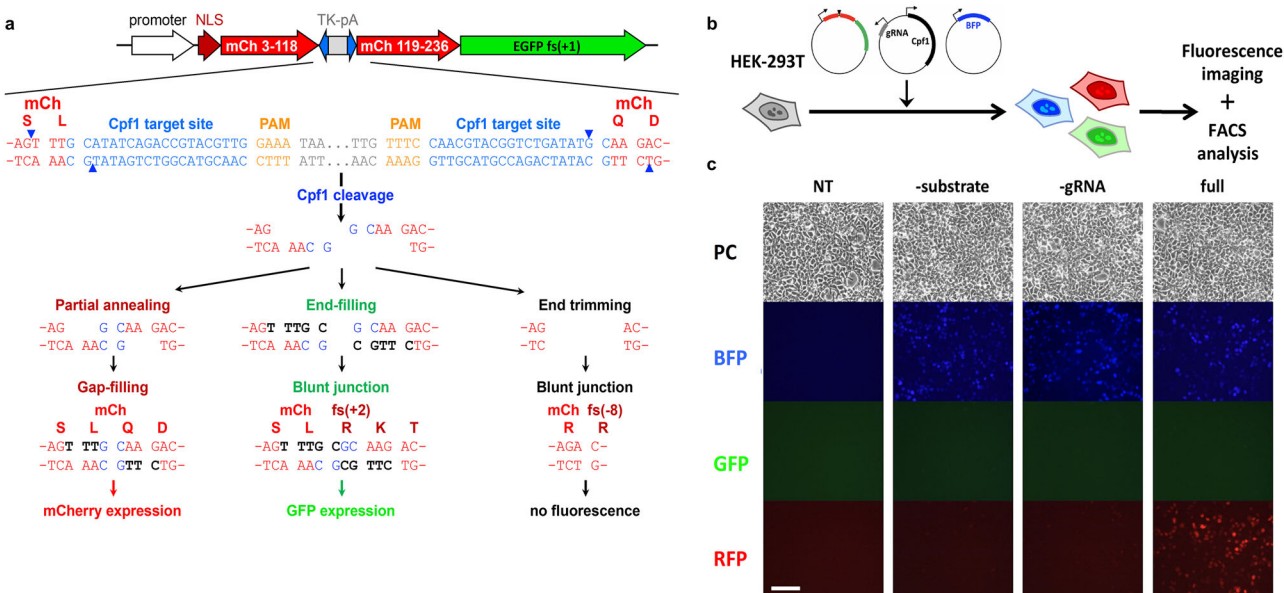

**Fig. 4 | Set up of a gap-filling assay in cells. a** Gap-filling reporter substrate. The gap-filling reporter substrate consists of two consecutive frameshifted (+1) coding sequences for mCherry (mCh) and EGFP, respectively. The mCherry coding sequence is interrupted by a cassette containing the HSV-TK polyadenylation sequence (TK-pA) flanked by two inverted copies of a Cpf1 target sequence (blue characters; the PAM GAAA sequence is shown in orange). Following Cpf1-mediated double cleavage, the HSV-TK polyadenylation sequence is deleted and the resulting 5' overhang DNA ends can be rejoined in different ways. First, the two overhangs can partially anneal to each other by two G:C pairs, leading to single-stranded gaps than can be filled, thus restoring an intact mCherry coding sequence. Second, the two overhangs can undergo end-filling, leading to blunt ends whose joining disrupts the mCherry reading frame (+2 frameshift) but enables the expression of the downstream EGFP coding sequence. Finally, the two overhangs can be trimmed off before end-joining, leading to a -8 frameshift of the mCherry coding sequence and resulting in neither red nor green fluorescence. **b** Gap-filling assay. The gap-filling reporter assay is performed by transfecting HEK-293T cells with the reporter substrate together with a Cpf1/gRNA-expressing vector to cleave the substrate and a BFP-expressing vector to normalize for transfection efficiency. Fluorescence expression is analyzed 48 h later by flow cytometry. **c** Representative fluorescence microscopy images of cells non-transfected (NT) or transfected with the full reporter system (full) or omitting the substrate (-substrate) or the Cpf1/gRNA expression vector (-gRNA). PC: phase contrast. Scale bar represents 100 μm. Source data are provided as a Source Data file.

---

mutated Ku70 forms (Supplementary Fig. 8a). Upon auxin addition, we analyzed in parallel the impact of individual mutations in Ku70 on gap-filling at staggered DSB and on direct EJ at blunt DSB (Fig. 5c). Compared to the slight reduction in gap-filling observed with Ku70 mutants at T307 and L310 positions, F303G mutation reduced gap-filling by 60% without impairing repair at blunt-ended breaks, indicating that Ku70 F303 is a crucial position for Ku interaction with Pol λ BRCT but not for Ku interaction with DNA. Similarly, we evaluated the impact of mutations of Ku80 by using HEK-293T cells expressing mAID-Ku80 fusion and transduced with WT or mutated Ku70 forms (Supplementary Fig. 8b). We found that mutations at Ku80 E292 and E304 positions reduced gap-filling and not direct end-joining (Fig. 5d).

### Pol μ binds to Ku70/80 at the bridge

Following our extensive investigation of Pol λ, we investigated whether Pol μ could bind Ku in a similar manner. First, we collected cryo-EM data on similar LRC samples, replacing Pol λ with Pol μ. However, we were unable to resolve extra density corresponding to Pol μ within the long-range complex maps. We therefore attempted to obtain structural data from a simplified system of only Ku70/80, DNA, and Pol μ, with specific microhomology DNA (see "methods")[25]. Within this cryo-EM dataset we resolved a map to 4.05 Å resolution, where we observed density for the BRCT domain of Pol μ (Supplementary Figs. 9, 10). In this map, we docked an AlphaFold model of Pol μ and following adjustment using namdinator[26] and manual refinements in coot[27] we were able to visualize the interface between Pol μ and Ku70/80 (Fig. 6a). Interestingly, although both Pol λ and Pol μ BRCT domains bind to the bridge of Ku70/80, their orientations are noticeably different (Fig. 6b, c). Pol λ is moved away from where DNA-PKcs binds, such that the position of Pol μ within Ku70/80 would clash with DNA-PKcs (Supplementary Fig. 11a).

Nevertheless, the Pol μ-Ku-DNA structure reveals that like for Pol λ, Ku80 E304 and Ku70 F303 are key positions for Pol μ interaction with the Ku70/80 bridge (Fig. 6a). In addition, it highlights a structural role of the conserved Pol μ R43 at this interface (Fig. 6a, Supplementary Fig. 11b). We then analyzed the impact of mutations in Pol μ BRCT on gap-filling by using Pol (λ, μ) double deficient HEK-293T cells complemented with WT or mutant Pol μ (Fig. 6d and Supplementary Fig. 8c). While Pol μ WT partially rescued the profound gap-filling defect of Pol (λ, μ) deficient cells, the R43A mutation significantly prevented this rescue. This validates the R43 position in Pol μ deduced from the cryo-EM structure above as key for Pol μ activity in NHEJ.

### Pol λ and Pol μ binding to Ku70/80 ensures survival to IR

Finally, we assessed the impact of impairing PolX:Ku interaction on cell radiosensitivity. Since we observed that the loss of cell viability following Ku removal is reversible within a 32 h time window (Supplementary Fig. 8d), this allowed us to assess the consequence of a Ku mutation at the Ku:Pol X interface on cell survival to ionizing radiation (IR) (Fig. 6e). We found that the Ku70 F303G mutant that preserved direct EJ but impaired gap-filling did not fully complement the radiosensitivity observed upon Ku removal. As we showed above, Pol μ partially compensated the absence of Pol λ on gap-filling activity and both activities were impaired by expression of Pol λ dead (Supplementary Fig. 7g, h and Fig. 6f). Notably, introducing the R57E mutation that impaired Pol λ recruitment to DSB in the Pol λ dead construct enhanced gap-filling activity back to the level observed in Pol λ KO cells (Supplementary Fig. 7g, h and Fig. 6f). Similarly, while cell resistance to IR was decreased upon expression of the Pol λ dead construct below that of Pol λ KO cells, the R57E in Pol λ dead construct reversed this sensitizing effect (Fig. 6g). These data support that Ku interaction with

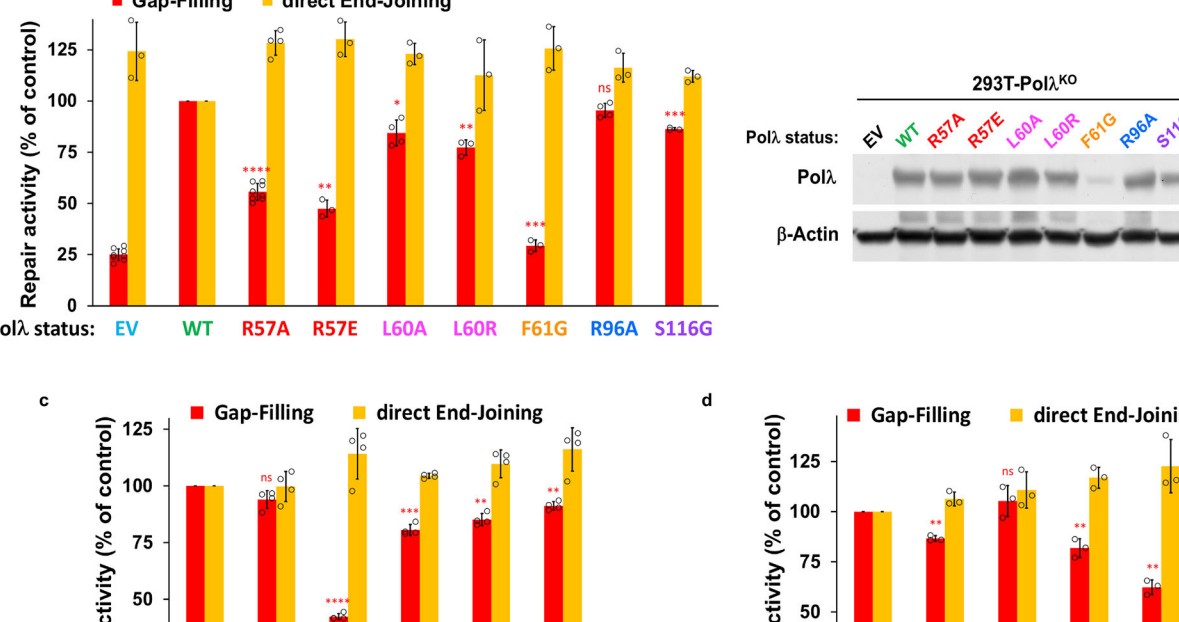

**Fig. 5 | Effect of Pol λ or Ku mutations on gap-filling activity in cells. a** Gap-filling activity (red bars) or direct end-joining activity (orange bars) assessed in parallel in HEK-293T cells knocked-out for Pol λ and complemented with either an empty vector (EV) or expression vectors for WT or mutant Pol λ. Results are normalized to the WT condition and plotted as mean values ± SD of 3–8 independent experiments (Gap-filling and direct End-Joining experiments, respectively: EV: n = 8 and 3; Polλ-WT: n = 12 and 9; -R57A: n = 8 and 4; -R57E: n = 3 and 3; L60A: n = 4 and 3; -L60R: n = 3 and 3; F61G: n = 3 and 3; -R96A: n = 4 and 3; -S116G: n = 3 and 3). Two-sided P-values from one-sample Student's *t*-test between the WT condition and the considered mutant, for the gap-filling assay, are as follows: R57A (<0.0001 ****), R57E (0.0021 **), L60A (0.0156 *), L60R (0.0089 **), F61G (0.0005 ***), R96A (0.0731 ns), S116G (0.0007 ***). * P < 0.05, ** P < 0.01, *** P < 0.001, **** P < 0.0001, ns: not significant. **b** Western blot on whole cell protein extracts from HEK-293T cells knocked-out (KO) for *POLL* and complemented with an empty vector (EV) or expression vectors for WT or mutant Pol λ. **c** Gap-filling activity (red bars) or direct end-joining activity

(orange bars) assessed in parallel in HEK-293T cells knocked-down for Ku70 and rescued with expression vectors for WT or mutant Ku70. Results are normalized to the WT condition and plotted as mean values of n = 4 independent experiments ±SD. Two-sided P-values from one-sample Student's *t*-test between the WT condition and the considered mutan*t*, for the gap-filling assay, are as follows: R301A (0.0541 ns), F303G (<0.0001 ****), T307A (0.0006 ***), L310G (0.0015 **), L310R (0.0025 **). **d** Gap-filling activity (red bars) or direct end-joining activity (orange bars) assessed in parallel in HEK-293T cells knocked-down for Ku80 and rescued with expression vectors for WT or mutant Ku70. Results are normalized to the WT condition and plotted as mean values of n = 3 independent experiments ± SD. Two-sided P-values from one-sample Student's *t*-test between the WT condition and the considered mutant, for the gap-filling assay, are as follows: E292A (0.0033 **), D301A (0.3587 ns), E304A (0.0049 **), E304R (0.0030 **). Individual values are shown as open circles. Source data are provided as a Source Data file.

gap-filling polymerases promotes cell survival to IR through positioning them at the break ends.

## Discussion

We determined the cryo-EM structures of Pol λ in complex with the Ku70/80 heterodimer bound to DNA whilst engaged with the DNA-PK holoenzyme. These structures show a clearly defined interaction between the BRCT domain of Pol λ and the interface between Ku70/80. We assessed the functionality of this interaction by generating specific site directed mutants on either Pol λ BRCT or Ku70/80. These mutants were used in two orthogonal assays: live protein recruitment at nuclear laser sites and gap-filling activity in cells transfected with an original reporter assay based on Cpf1-generated partially complementary DSB ends. These data define the molecular basis and essentiality of the BRCT domain for recruitment of Pol λ within the NHEJ complex. The data also establish key positions on Ku70/80 and Pol λ-BRCT that mediate their interaction, and for the first time position the interaction region at the external face of the Ku70/80 dimer interface.

From our cryo-EM structure of the Ku80 mediated DNA-PK dimer bound to LX4, the LigIV tandem BRCT1 domain can be seen occupying the previously described site on Ku70/80[6,7] that is distinct from the Pol

λ BRCT interaction site (Figs. 1 and 6). The LigIV BRCT1 sits in the groove formed by the Ku70/80 dimer interface, whereas Pol λ BRCT is located at an adjacent site. From our structural data, it is apparent that the two proteins have distinct sites of contact with Ku70/80, and there is no evidence that Pol λ and LigIV form direct interactions with each other. Thus, the previously reported observation that specific Pol λ mutations prevent complex formation with Ku70/80-LigIV-XRCC4 are likely due to disruption of the Pol λ:Ku70/80 interaction specifically, as has previously been suggested[16]. Interestingly, when we collected cryo-EM data of Ku70/80 alone (without DNA-PKcs) with Pol λ, we did not observe any density for the BRCT domain of Pol λ engaged with Ku70/80. Therefore, it is possible that the Pol λ interaction with Ku70/80 is stabilized within the DNA-PK holoenzyme complex. Our cryo-EM structures indicate a possible weak interaction between the BRCT domain of Pol λ and DNA-PKcs. Although the resolution is low, we would predict that Pol λ (residues 128-133) may interact with residues Ser187 and/or Glu188 on DNA-PKcs (Supplementary Fig. 11c).

Notably, we found that recruitment of the N-terminal Pol λ BRCT domain to micro-irradiated areas was essentially Ku70/80-dependent in contrast with data obtained with the full-length Pol λ. This suggests that Pol λ recruitment to sites of DNA damage may rely on interactions

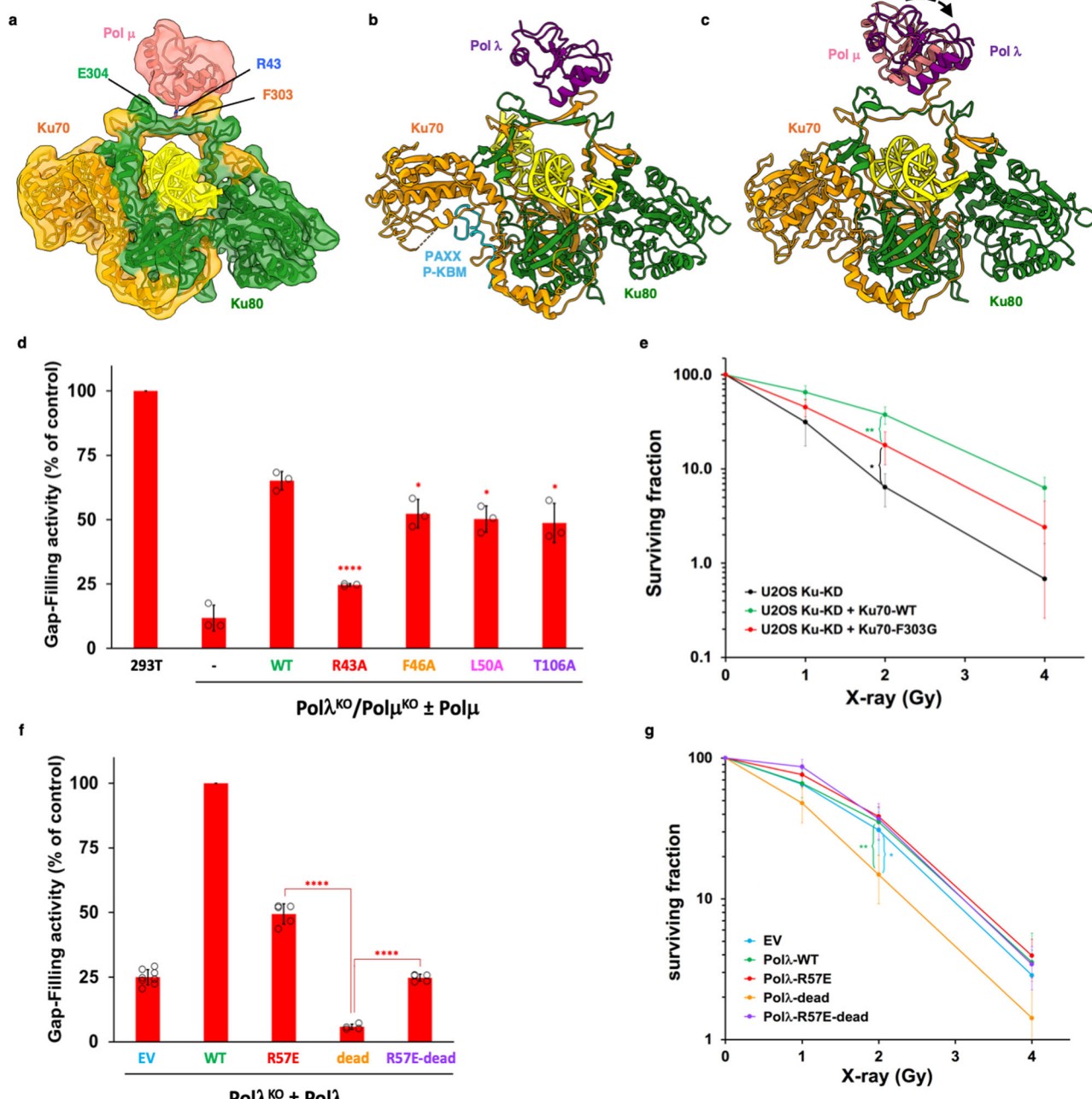

**Fig. 6 | Comparison of Cryo-EM structures of Pol λ and Pol μ on Ku70/80 and functional assays with polymerases or Ku mutants. a** Cryo-EM structure of Ku70/80 bound to the BRCT domain of Pol μ at 4.05 Å resolution. Ku70 in orange, Ku80 in green, DNA in yellow and Pol μ in coral. **b** Ku70/80 bound to the BRCT domain of Pol λ from the LRC in Fig. 1. Pol λ in purple and the PAXX P-KBM in blue. **c** Overlay of the structure of Pol μ bound to Ku70/80 from A with Pol λ. **d** Gap-filling activity assessed in HEK-293T cells knocked-out for *POLL* and *POLM* and complemented with ectopic expression of wild-type (WT) or mutant Pol μ. Results are normalized to the WT condition (parental cells) and plotted as mean values of n = 3 independent experiments ± SD. Two-sided P-values from unpaired two-sample *t*-test between the WT condition and the considered mutant are as follows: R43A (<0.0001 ****), F46A (0.0282 *), L50A (0.0147 *), T106A (0.0285 *). Individual values are shown as open circles. **e** Cell survival to X-rays of U2OS cells knocked-down for Ku70 (Ku-KD) and complemented or not with expression vectors for WT Ku70 or the F303G mutant (see Fig. 3b). Results are mean values of n = 4 independent experiments. Two-sided P-values were calculated at 2 Gy using an unpaired two-

sample *t*-test: Ku70-F303G versus Ku-KD (0.0195 *); Ku70-F303G versus Ku-WT (0.0089 **). **f** Gap-filling activity assessed in HEK-293T cells knocked-out for *POLL* and complemented with ectopic expression of WT or mutant Pol λ (see Fig. S7G). Results are normalized to the WT condition and plotted as mean values of 4 to 8 independent experiments ±SD (EV: n = 8; Polλ-WT: n = 10; -R57E: n = 5; -dead: n = 4; -R57E-dead: n = 5). Two-sided P-values from unpaired two-sample *t*-test between the considered mutants are as follows: Polλ-dead versus Polλ-R57E (<0.0001 ****), Polλ-dead versus Polλ-R57E-dead (<0.0001 ****). Individual values are shown as open circles. **g** Cell survival to X-rays of HEK-293T cells knocked-out for *POLL* and complemented with an empty vector (EV) or expression vectors for WT or mutant Pol λ (see Fig. 6F). Y axis is log scale. Results are normalized to the untreated condition and plotted as mean values of 4–7 independent experiments ±SD (EV: n = 7; Polλ-WT: n = 4; -R57E: n = 5; -dead: n = 5; -R57E-dead: n = 6). Two-sided P-values were calculated at 2 Gy using an unpaired two-sample *t*-test: Polλ-dead versus EV (0.0207 *); Polλ-dead versus WT (0.0092 **). Source data are provided as a Source Data file.

outside of its BRCT domain. This may reflect additional roles of Pol λ outside of classical NHEJ such as in Base Excision Repair where it is proposed to interact with DNA glycosylases involved in the repair of alkylated or oxidized bases[20]. As previously reported, we found that Pol λ deficient cell lines showed modest, if any, sensitivity to IR, whereas cells deficient in both λ and μ polymerases were clearly IR sensitive[14,28,29]. The expression of a catalytically inactive form of Pol λ concurrently inhibited gap-filling activity in cells (this work) and negatively impacted on cell survival to IR as previously reported[24], to extents higher than the sole deletion of Pol λ. This suggests that Pol λ dead prevents the rescue of the Pol λ defect by Pol μ, likely through occupying and occluding a common interaction site on Ku70/80, and thereby sustaining a dominant negative effect. Also, we showed that Ku70 F303G mutant cells exhibit sensitivity to IR equivalent to that of a double Pol λ-Pol μ KO cells, suggesting again that this mutation impairs Ku70/80 interaction with both Pol X proteins. Indeed, the structures of Pol λ and Pol μ BRCTs with Ku70/80 that we report here, together with AlphaFold prediction of the structure formed between Ku70/80 with DNA and the TdT BRCT domain indicate the use of similar protein:protein interfaces with the common Ku80 E304 and Ku70 F303 key positions (Supplementary Fig. 12a–c). Notably, the broad composition and organization of the Pol λ BRCT domain are conserved amongst Pol X members, supporting a conserved mode of interaction with Ku70/80 (Supplementary Fig. 11b). Together with our present cellular data, this allows us to propose a unified model in which the three BRCT-bearing Pol X interact with Ku70/80 on the same site, allowing their respective enrollment in the NHEJ complex.

The Ku70/80 interaction with Pol X BRCT domain reported here adds to the fascinating list of Ku70/80 sites that contact NHEJ factors. These include Ku80 sites for A-KBM bearing proteins (APLF, WRN, MRI), XLF via its X-KBM[30] and DNA-PKcs[8], Ku70 sites for PAXX via its P-KBM[2] and LigIV via its BRCT1[7,8], and combined sites in Ku80/Ku70 for inositol hexaphosphate[19] and now for Pol X BRCTs (B-KBM). The structures reported here of the Ku70/80:Pol λ interaction represents a paradigm for Pol X mode of binding to Ku70/80 and reinforces the notion that Ku70/80 acts as a central structural hub in the NHEJ mechanism[31]. Like other Ku70/80 partners whose KBM motifs are at the N or C-terminus of the protein, Pol X members are anchored by an N-terminal BRCT domain separated by a flexible region from the catalytic portion of the protein, allowing free movement for the correct positioning at the site of the DSB.

The structures of the LRC presented here indicate that the Pol λ:Ku70/80 interaction occurs early during the NHEJ process, at a stage where DNA-PKcs is still present and at which the DNA ends are likely not yet accessible to processing enzymes. Moreover, since our cryo-EM LRC structures with Pol λ were obtained using a blunt-ended DNA substrate, this suggests that it is not the nature of the DNA ends that dictates Pol X engagement into the NHEJ process but rather the intrinsic affinity for the initial DNA-PK complex. These structures also illustrate that the evolution of distinct and independent binding sites on the Ku70/80 heterodimer enables the concurrent anchoring of NHEJ proteins (PAXX/XLF, DNA-PKcs, Pol X, XRCC4/LigIV) dedicated to specific activities involved in the repair reaction (synapsis, kinase, polymerase, ligase, respectively) (Supplementary Fig. 12d). Moreover, the dimeric nature of the LRC allows for a scenario where two different members of Pol X family could be engaged within the same dimeric complex simultaneously. Although our DSB repair assay in cells could theoretically report both gap-filling and end-filling activities, we found that gap-filling is far dominant. This suggests that, when possible, end-annealing on minimal homology is preferred, most likely because it contributes to the stability of the synaptic complex and ultimately to the fidelity of DNA repair. The two-nucleotide overhang substrate used here supports a tight coupling between end annealing and gap filling, as demonstrated elegantly for NHEJ mediated repair in Xenopus egg extracts[23]. Within the NHEJ LRC, this coupling is ensured by the concurrent binding by Ku70/80 of DNA ends and resynthesis factors on both sides of the DSB. Also, the coexistence of LigIV and processing

enzymes like Pol X in the synaptic complex at break ends ensures that ligation can proceed as soon a DNA ends are ligatable, limiting unnecessary DNA sequence alteration[23]. This illustrates that the unique capability of Ku70/80 for simultaneous multivalent interaction is crucial for the high adaptability of the NHEJ process[32].

## Methods

### Purification of DNA-PKcs
DNA-PKcs was isolated from HeLa cells. All purification steps were carried out at 4 °C to ensure protein stability. Frozen nuclear extracts were initially dialyzed into buffer A (20 mM HEPES, pH 7.6, 100 mM NaCl, 10% glycerol (vol/vol), 0.5 mM EDTA, 2 mM MgCl$_2$, 5 mM DTT, 0.2 mM PMSF) and the protein was isolated using several ion-exchange columns: Q sepharose, HiTrap heparin, Mono-S and Mono-Q eluted with a NaCl gradient of 0.1–1 M using buffer B (buffer A but with 1 M NaCl). As a final polishing step, the protein was applied to a Superose 6 size exclusion chromatography column equilibrated with buffer C (20 mM HEPES, pH 7.6, 200 mM NaCl, 10% glycerol (vol/vol), 0.5 mM EDTA, 2 mM MgCl$_2$, 5 mM DTT). SDS–PAGE was carried out during purification with a DNA-PKcs control to indicate sample purity. The glycerol percentage was adjusted to 30% (vol/vol) and the protein flash-cooled in liquid nitrogen before being stored at −80 °C until further use.

### Purification of Ku70/80
A pFASTBac-dual expression vector encoding full-length His-tagged Ku70/80 was expressed in Sf9 insect cells. All purification steps were carried out at 4 °C to ensure protein stability. Following expression, cell pellets were resuspended in lysis buffer (50 mM Tris, pH 8.0, 5% glycerol (vol/vol), 150 mM NaCl, 2 mM β-mercaptoethanol, 20 mM imidazole, 10 protein inhibitor cocktail tablets, 20 mg ml$^{-1}$ deoxyribonuclease I) and the cells were sonicated. The resulting lysate was then mixed for 30 min with the addition of NaCl (final concentration 500 mM) and centrifuged (45 min, 30,000 × g, 4 °C). The supernatant was purified using Ni-NTA resin (Qiagen) previously equilibrated with binding buffer (50 mM Tris, pH 8.0, 5% glycerol (vol/vol), 500 mM NaCl, 2 mM β-mercaptoethanol, 20 mM imidazole) and eluted using buffers containing 100 mM and 300 mM imidazole. Eluted Ku70/80 was bound to a Q sepharose anion exchange column in buffer A (20 mM Tris, pH 8.0, 50 mM NaCl, 5% glycerol (vol/vol), 5 mM DTT) and eluted using a linear gradient of buffer A with 1 M NaCl. Finally, the protein was applied to an S200 Superdex size exclusion column (GE Healthcare) equilibrated with 50 mM Tris pH 8.0, 150 mM, 5% glycerol (vol/vol) and 5 mM DTT. SDS–PAGE gel analysis was used to assess protein purity and the proteins were concentrated and flash-cooled in liquid nitrogen before being stored at −80 °C for further use.

### Purification of LX4
Expression and purification of Lig4/XRCC4 (LX4) were carried out according to ref. [8]. Briefly, a construct containing full-length 10xHis-tagged LX4 were expressed in insect cells. Following expression, cell pellets were resuspended in lysis buffer [20 mM tris (pH 8.0), 5% glycerol, 50 mM KCl, 50 mM NaCl, 5 mM β-mercaptoethanol, 25 mM imidazole, and 2 protein inhibitor cocktail tablets per liter], and cells were sonicated. The resulting lysate was then mixed with 2 μl of benzonase (25 kU of stock, activity per microliter, origin) and MgCl$_2$ to a final concentration of 5 mM and left on ice for 20 min. The lysate was then centrifuged (30,000 × g for 20 min at 4 °C). The supernatant was purified using Ni–nitrilotriacetic acid resin (QIAGEN) previously equilibrated with lysis buffer and eluted using the lysis buffer containing 300 mM imidazole. Eluted XLF was bound to a Resource Q sepharose anion exchange column in buffer A [20 mM tris (pH 8.0), 50 mM KCl, 50 mM NaCl, 5 mM β-mercaptoethanol, and 1 mM EDTA] and eluted using a linear gradient of buffer A with 850 mM NaCl. Last, the protein was dialyzed into a final buffer of 10 mM tris (pH 8.0), 150 mM NaCl, and 5 mM β-mercaptoethanol before being stored at −80 °C for further use.

## Purification of PAXX

Full-length PAXX was expressed and purified according to Ochi et al.[33]. Briefly, PAXX was cloned into a pHAT4 vector and optimized for expression in *Escherichia coli*. The protein was expressed in BL21(DE3) cells and purified with a 6xHis-tag by Ni-affinity chromatography. The protein was then purified further using cation exchange and size exclusion chromatography.

## Polymerase λ expression and purification

The DNA construct for DNA polymerase λ was ordered as a gene inserted into a pET28a plasmid which already contains a N-terminal 6x His-tag (BaseClear). The plasmid was transformed into Rosetta 2(DE3) cells. The cells were grown in LB media and protein expression was induced once the $OD_{600}$ reached 0.6 with 0.5 mM IPTG, and then left to grow overnight at 16 °C, 180 rpm. The cells were harvested by centrifugation ($4000 \times g$, 30 min, 4 °C) and resuspended in lysis buffer (20 mM Tris pH 8.0, 300 mM NaCl, 10% glycerol (w/v), 1 mM β-mercaptoethanol, 1 protease inhibitor cocktail tablet) prior to sonication. The cell debris was pelleted by centrifugation ($18,000 \times g$, 30 min, 4 °C), and the supernatant was loaded onto a 5 ml HisTrap NI-NTA HP column equilibrated with buffer A (20 mM Tris pH 8.0, 300 mM NaCl, 10% glycerol, 1 mM β-mercaptoethanol). The protein was eluted using a gradient with buffer A containing 500 mM imidazole. Eluted polymerase was loaded onto a 5 ml HiTrap Q column equilibrated in buffer B (20 mM Tris pH 9.2, 50 mM NaCl, 10% glycerol, 5 mM DTT), following overnight dialysis at 4 °C with agitation in 2 L of buffer B. Gradient elution was carried out with buffer B containing 1 M NaCl. The fractions containing polymerase λ were pooled and concentrated using centricon (Amicon) with a 10 kDa cut-off. The protein was injected onto a Superdex 200 increase 10/300 GL column which had been equilibrated in buffer C (20 mM Tris pH 8.0, 10% glycerol, 150 mM NaCl, 5 mM DTT). The fractions corresponding to the protein were pooled and buffer exchanged into cryo-EM buffer (20 mM Hepes pH 7.6, 200 mM NaCl, 0.5 mM EDTA, 2 mM $MgCl_2$, 5 mM DTT).

## Polymerase μ expression and purification

The DNA construct for DNA polymerase μ was ordered as a gene inserted into a pET28a plasmid which already contains a N-terminal 6x His-tag (BaseClear). The plasmid was transformed into Rosetta 2(DE3) cells. The cells were grown in LB media and protein expression was induced once the $OD_{600}$ reached 0.6 with 1 mM IPTG, and then left to grow overnight at 16 °C, 180 rpm. The cells were harvested by centrifugation ($4000 \times g$, 30 min, 4 °C) and resuspended in lysis buffer (20 mM Hepes pH 7.0, 300 mM NaCl, 10% glycerol (w/v), 1 mM β-mercaptoethanol, 1 protease inhibitor cocktail tablet) prior to sonication. The cell debris was pelleted by centrifugation ($18,000 \times g$, 30 min, 4 °C), and the supernatant was loaded onto a 5 ml HisTrap NI-NTA HP column equilibrated with buffer C (20 mM Hepes pH 7.0, 300 mM NaCl, 10% glycerol, 1 mM β-mercaptoethanol). The protein was eluted using a gradient with buffer C containing 500 mM imidazole. Eluted polymerase was concentrated using centricon (Amicon) with a 30 kDa cut-off. The protein was injected onto a Superdex 200 increase 10/300 GL column which had been equilibrated in buffer D (20 mM Hepes pH 7.0, 10% (w/v) glycerol, 150 mM NaCl, 5 mM DTT). The fractions corresponding to the protein were pooled and buffer exchanged into cryo-EM buffer (20 mM Hepes pH 7.6, 200 mM NaCl, 0.5 mM EDTA, 2 mM $MgCl_2$, 5 mM DTT).

## Electromobility shift assays

Electromobility shift assay (EMSAs) were used to determine what ratio of polymerase is required to stabilize binding to Ku70/80. The reactions were pre-formed in a final volume of 20 μl containing cryo-EM buffer, 5% (w/v) glycerol, 50 nM Y-shape DNA, 200 nM Ku and different concentrations of the polymerases. The samples were incubated for 10 min at 4 °C and then ran on TBE Gels 4–12% with 1X TBE buffer at 150 V for 45 min. The gels were stained with SYBR™ Gold Nucleic Acid Gel stain (Thermo Fisher Scientific) as per manufacturer's recommendation and visualized.

## Expression and purification of full-length XLF and LX4

A construct containing full-length 10xHis-tagged XLF and LX4 (LigIV and XRCC4) were expressed in insect cells and purified as previously reported[2].

**DNA annealing.** Biotinylated Y-shaped 42-55 bp dsDNA and microhomology DNA were synthesized and annealed as described previously[6,25]. DNA sequences used for annealing can be found below.

Y-shaped DNA Forward

Biotin-CGCGCCCAGCTTTCCCAGCTAATAAACTAAAAACTATTATTATGGCCGCACGCGT

Y-shaped DNA Reverse

ACGCGTGCGGCCATAATAATAGTTTTTAGTTTATTGGGCGCG

Microhomology DNA strands for Pol μ

5′ – GGTGCAGCACAGTGC – 3′

5′ – GCCGTGCAGTC – 3′

5′- GACTGCACGGCAGC -3′

5′- ACTGTGCTGCACC -3′

## Cryo-EM sample preparation of DNA-PK, LX4, PAXX, and Polymerase λ structure

Proteins were concentrated using a centrifugal filter (Amicon) with a 30 kDa cut-off and buffer exchanged into 20 mM HEPES, pH 7.6, 200 mM NaCl, 0.5 mM EDTA, 2 mM $MgCl_2$, 5 mM DTT. Purified Ku70/80 full-length was then first mixed with Y-shaped 42–55 bp DNA before being mixed with purified DNA-PKcs, LX4, PAXX, and Pol λ in respectively a 1:1:2:2:2:6 (DNA:DNA-PKcs:Ku:LX4:PAXX:Polλ) ratio.

## Cryo-EM grid preparation Pol λ

Aliquots of 3 μl of ~2.5 mg/ml of the NHEJ complex (based on DNA-PKcs concentration) were mixed with 8 mM CHAPSO to eliminate particle orientation bias (final concentration; Sigma) before being applied to Holey Carbon grids (Quantifoil Cu R1.2/1.3, 300 mesh), glow discharged for 60 s at current of 25 mA in PELCO Easiglow (Ted Pella, Inc.). The grids were then blotted with filter paper once to remove any excess sample, and plunge-frozen in liquid ethane using a FEI Vitrobot Mark IV (Thermo Fisher Scientific) at 4 °C and 95% humidity.

## Cryo-EM sample preparation of Ku70/80-DNA and Pol μ

Purified Ku70/80 full-length was then first mixed with microhomology DNA (sequence given above) and then with full-length Pol μ in a ratio of 1:2:6 of DNA:Ku:Pol μ (established from EMSA studies (Supplementary Fig. 1)).

## Cryo-EM grid preparation for Ku70/80, DNA, and Pol μ

Aliquots of 3 μl of ~3 mg/ml of the Ku70/80-Pol μ complex were mixed with 8 mM CHAPSO to eliminate particle orientation bias (final concentration; Sigma) before being applied to Holey Carbon grids (Quantifoil Cu R1.2/1.3, 300 mesh), glow discharged for 60 s at current of 25 mA in PELCO Easiglow (Ted Pella, Inc.). The grids were then blotted with filter paper once to remove any excess sample, and plunge-frozen in liquid ethane using a FEI Vitrobot Mark IV (Thermo Fisher Scientific) at 4 °C and 95% humidity.

## Cryo-EM data acquisition

The data was collected on a Titan Krios equipped with a Gatan K3 direct electron counting detector at the University of Leicester. All data collection parameters are given in Supplementary Table 1.

## Cryo-EM Image processing

The classification process for the datasets is summarized schematically in Supplementary Figs. 2 and 9. The final reconstructions obtained had overall resolutions (Supplementary Table 1), which were calculated by Fourier shell correlation at 0.143 cut-off.

## Cryo-EM structure refinement and model building

The model of the DNA-PK monomer (PDB:7NFE) was used as an initial template and rigid-body fitted into the cryo-EM density in UCSF chimera[34] and manually adjusted and rebuilt in Coot[27]. LX4 was either removed or manually adjusted depending on whether density was present and PAXX was docked within the central density. The BRCT domain of Polymerase λ was then manually docked into the density (PDB: 2JW5) in chimera. Models and their corresponding density maps were run through Namdinator[26] before being refined using Phenix real-space refinement[35].

## Cell lines, cell culture, and cell engineering

U2OS cells (human osteosarcoma cell line from ECACC, Salisbury, UK) and HEK-293T human embryonic cells, were grown in DMEM (Eurobio, France) supplemented with 10% fetal calf serum (Eurobio, France), 125 U/ml penicillin, and 125 µg/ml streptomycin. Cells were maintained at 37 °C in a 5% $CO_2$ humidified incubator.

HEK-293T cells knocked-out for *POLL* (DNA Polymerase λ), *POLM* (DNA Polymerase µ), PRKDC (DNA-PKcs, Addgene Plasmid#220493[36]), LigIV, XLF, PAXX and XRCC5 (Ku80) genes were obtained following cell transfection with the pCAG-eCas9-GFP-U6 vector expressing the corresponding guide RNA (see below Supplementary Table 2) using jetPEI (Polyplus) as a transfection reagent. Following cell sorting, individual clones were isolated and checked by western blot.

The generation of U2OS cells expressing an inducible shRNA against Ku80 and the generation of U2OS and HEK-293T cells expressing a mini-auxin-inducible degron-tagged Ku70 protein (mAID-Ku70) in place of the endogenous Ku70 protein have been previously described[2,19]. HEK-293T cells expressing an auxin-degradable mAID-Ku80 construct in place of the endogenous Ku80 protein were obtained, first, by transducing the cells with two lentiviral vectors enabling expression of mAID-Ku80 and a TIR1 construct, and second, by knocking-out the endogenous XRCC5 gene, using previously described materials and protocols[19].

Production of lentiviral particles in HEK-293T cells and transduction of U2OS and HEK-293T cells were performed as previously described[37].

## Western-blot and antibodies

Cell pellets were washed with phosphate-buffered saline (PBS) and resuspended in lysis buffer (50 mM Hepes-KOH, pH 7.5, 450 mM NaCl, 1 mM EDTA, 1% Triton X-100) supplemented with Halt protease inhibitor cocktail (ThermoFisher Scientific). Cells were lysed by four freeze/thaw cycles in liquid nitrogen and 37 °C water bath. Lysates were cleared by centrifugation and protein concentrations were determined using the Bradford assay (Bio-Rad, Hercules, CA). Equal amounts of proteins were mixed with concentrated loading sample buffer to 1X final concentration (50 mM Tris.HCl pH 6.8, 10% glycerol, 1% SDS, 300 mM 2-mercaptoethanol, 0.01% bromophenol blue), heat-denatured, separated by SDS-PAGE on Miniprotean TGX stain-free 4–15% gradient gels (Bio-Rad, Hercules, CA) and blotted onto Protran 0.45 µm nitrocellulose membranes (GE Healthcare). Membranes were blocked for 60 min with 5% non-fat dry milk in PBS, 0.1% Tween-20 (Sigma-Aldrich) (PBS-T buffer), incubated as necessary with primary antibody diluted 1/1000 in PBS-T containing 1% bovine serum albumin (immunoglobulin- and lipid-free fraction V; Sigma-Aldrich) and washed 3 times with PBS-T. Membranes were incubated for 1 h with HRP-conjugated secondary antibodies (Jackson Immunoresearch Laboratories) diluted 1/5000 in PBS-T and washed five times with PBS-T.

Immuno-blots were visualized by enhanced chemiluminescence (Western Lightning Plus-ECL; Perkin Elmer) and autoradiography. Primary antibodies used: mouse monoclonal antibodies anti-DNA-PKcs (clone 18.2; Thermo Fisher Scientific, MA5-13238, lot: Y14047369), anti-Ku80 (clone 111; Thermo Fisher Scientific, MA5-12933, lot: WB31922872), anti-Ku70 (clone N3H10; Thermo Fisher Scientific, MA5-13110, lot: WE27669780), anti-Pol lambda (clone E11; Santa Cruz, sc-373844, lot: B0818), anti-beta-Actin (clone C4; Santa Cruz, sc-47778, lot: A2023); rabbit monoclonal antibodies anti-LigIV (Abclonal, A11432, lot: 4000000595) and anti-Pol mu (Abcam, EPR10470(B), lot: GR117969-3); rabbit polyclonal antibodies anti-XLF (Abclonal, A19957, lot: 00040701118) and anti-PAXX (Novus, NBP1-94172, lot: C118006).

## Ionizing irradiation and cell survival analysis

Two to six thousand U2OS or HEK-393T cells per well were seeded in duplicate in six-well plates. Cells were exposed 24 h later to various doses of X-ray using a Faxitron RX-650 device (130 kV, 5 mA, dose rate 0.3 Gy per min). Six to seven days later, cells were fixed with 7% trichloroacetic acid for 1 h at 4 °C. Fixed cells were extensively washed with water and plates were air dried before staining for 15 min with crystal violet (0.1% aqueous solution). Stained cells were further extensively washed with water and plates were air dried. Staining was dissolved with 10% acetic acid solution and absorption was measured at 570 nm (Ultrospec-3000 spectrophotometer, Pharmacia Biotech). Results were plotted as mean values of n > 3 independent experiments ±s.d. using Microsoft Excel software.

## Multiphoton laser micro-irradiation

Live cell microscopy and multiphoton laser micro-irradiation were conducted as previously described[30].

## In vivo DNA end-joining assays

To assess gap-filling activity, HEK-293T cells were seeded to 20–40% confluence in 6-well plates and transfected 24 h later with a mix of Cpf1-targeted gap-filling reporter substrate, Cpf1/gRNA expressing vector and mTagBFP2 expressing plasmid as an internal control. Cells were trypsinized 2 days post-transfection, washed with PBS and analyzed by flow cytometry on a Fortessa X-20 cell analyzer (BD Biosciences). The integrated red fluorescence signal accounting for gap-filling-mediated repair events (% positive cells x mean fluorescence) was normalized to that of transfection efficiency (BFP) (Supplementary Fig. S13). For the repair junction analysis, a variant of the substrate vector was generated in which a sequence encoding HygroR-T2A was inserted in frame, upstream of the disrupted mCherry sequence. This plasmid was stably transfected into HEK-293T cells and, after selection with hygromycin, positive clones were isolated. Following further transient transfection with the Cpf1/gRNA expressing vector, red fluorescent and non-fluorescent clones were isolated. Sequence junctions were analyzed by genomic DNA extraction (SV Genomic DNA Purification System, Promega), PCR amplification with primers mCh-Xba-F and mCh-Xho-R and sequencing (Eurofins Genomics; Ebersberg, Germany).

Direct end-joining activity was assessed as described previously with a dedicated reporter substrate in which blunt-ended DSBs were generated with appropriate Cas9/gRNA co-expression[19].

## Plasmids and DNA manipulations

Cas9 and gRNA expressing vectors for gene knockout were generated by inserting the pre-annealed gRNA-GOI-F and gRNA-GOI-R oligonucleotides of the corresponding targeted gene of interest into the BbsI restriction sites of pCAG-eCas9-GFP-U6-gRNA plasmid (a gift from Jizhong Zou, Addgene plasmid # 79145; http://n2t.net/addgene:79145; RRID:Addgene_79145). See below the list of oligonucleotides.

The previously described pLV3 lentiviral vector[30] was modified as follows to allow insertion of Pol λ cDNA downstream a puromycin

resistance gene and a sequence encoding the T2A ribosomal skipping peptide: first, a T2A cassette (pre-annealed oligonucleotides kpn2-T2A-Mlu-F and kpn2-T2A-Mlu-R) was inserted into the Kpn2I and MluI restriction sites of the pLV3 plasmid; second, a PCR-amplified fragment with a Puro-resistance cDNA sequence (PCR reaction with primers HF-Puro-F and HF-Puro-R on a synthetic DNA molecule as a template) was added by Hot-Fusion[38] at the Kpn2I site, resulting in the pLV3-Puro-T2A plasmid. A PCR-amplified human Pol λ cDNA fragment (primers PolL-Mlu-F and PolL-Bcu-R) was then inserted between the MluI and BcuI restriction sites of pLV3-Puro-T2A.

Expression vectors for mutant forms of Pol λ were obtained in a similar manner following an additional step of overlap extension PCR mutagenesis with the corresponding PolL-mut-F and PolL-mut-R oligonucleotides as mutated inner primers (see below the list of primers).

The expression vector for GFP-tagged Pol λ constructs (full-length protein or BRCT domain) was obtained by replacing the FLAG-Ku70 cDNA from the previously described pLV3-GFP-FLAG-Ku70 vector[2] by a linker cassette (pre-annealed Kpn2-AX-Mlu-F and Kpn2-AX-Mlu-R oligonucleotides) between the Kpn2I and MluI sites. The resulting pLV3-GFP plasmid was then used to insert between the MluI and BcuI sites the PCR-amplified human Pol λ cDNA fragment described above. Expression vectors for GFP-tagged WT and mutant Pol λ BRCT domain (aminoacids 1-136) were obtained in a similar manner following a PCR amplification of the corresponding cDNAs with PolL-Mlu-F and PolL-BRCT-Bcu-R primers and full-length Pol λ expression vectors as templates.

The expression vector for wild-type Pol μ was obtained by PCR amplification of the human Pol μ cDNA with primers PolM-Mlu-F and PolM-Bcu-R followed by Hot-Fusion insertion between the MluI and BcuI restriction sites of pLV3-Puro-T2A.

Expression vectors for mutant forms of Pol μ (R43A, F46A, L50A, and T106A) were obtained in a similar manner following an additional step of overlap extension PCR mutagenesis with the corresponding PolM-mut-F and PolM-mut-R oligonucleotides as mutated inner primers (see below the list of primers).

The lentiviral vector allowing expression of mCherry-tagged human PAXX was obtained, first, by inserting between Kpn2I and MluI restriction sites of pLV3 the coding sequence of mCherry, following PCR amplification from the pmCherry-NLS plasmid (a gift from Martin Offterdinger; Addgene plasmid # 39319; http://n2t.net/addgene:39319; RRID:Addgene_39319) with primers mCh-Kpn2-F and mCh-Mlu-R, second, by further inserting between MluI and BcuI restriction sites the coding sequence of PAXX, following PCR amplification from the previously described pLV3-GFP-PAXX plasmid[2] with primers PAXX-Mlu-F and PAXX-Bcu-R.

Ku70 lentiviral expression vector was generated by PCR amplification of human Ku70 cDNA with Ku70-Kpn2-F and Ku70-Bcu-R primers and subsequent insertion between Kpn2I and BcuI restrictions sites of the pLV3 vector. Expression vectors for mutant forms of Ku70 were obtained in a similar manner following an additional step of overlap extension PCR mutagenesis with the corresponding Ku70-mut-F and Ku70-mut-R oligonucleotides as mutated inner primers (see below the list of primers).

Ku80 lentiviral expression vector was already described[19]. Expression vectors for mutant forms of Ku80 were generated following an additional step of overlap extension PCR mutagenesis with the corresponding Ku80-mut-F and Ku80-mut-R oligonucleotides as mutated inner primers (see below the list of primers).

The gap-filling reporter substrate was assembled into the pEGFP-N1 vector (Clontech) following sequential insertion of various PCR products, oligonucleotide linkers and synthetic DNA fragments (see Fig. 4a for a detailed description). The Cpf1 and guide RNA co-expression plasmid used to cleave the reporter substrate was generated by inserting the pre-annealed gRNA-GF-F and gRNA-GF-R oligonucleotides into the Esp3I restriction sites of pTE4398 (a gift from

Ervin Welker; Addgene plasmid # 74042; http://n2t.net/addgene:74042; RRID:Addgene_74042).

The pNLS-mTagBFP2 plasmid used as an internal control of transfection efficiency was obtained by PCR amplification of the 2xNLS-mTagBFP2 coding sequence with mTagBFP-Acc65-F and mTagBFP-Mlu-R primers on the pHAGE-TO-nls-st1dCas9-3nls-3XTagBFP2 plasmid template (a gift from Thoru Pederson; Addgene plasmid # 64512; http://n2t.net/addgene:64512; RRID:Addgene_64512). The resulting PCR fragment was inserted into the pEGFP-N1 (Clontech) vector backbone after modification of the multiple cloning site and removal of the GFP coding sequence.

All oligonucleotides were purchased from Eurofins Genomics (Ebersberg, Germany). Restriction and modifying enzymes (Phusion and T4 DNA Ligase) were from ThermoFisher

Scientific (Illkirch, France). All constructs were checked by sequencing (Eurofins Genomics).

## Reporting summary

Further information on research design is available in the Nature Portfolio Reporting Summary linked to this article.

## Data availability

All structural data presented are publicly available. Cryo-EM structures and maps are deposited at the PDB and EMDB with accession codes as follows: 9GD7, 9G9L and 9IO4 and EMD: 51249, 51156, and 52550. Source data are provided as a Source Data file. Source data are provided with this paper.

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

## Acknowledgements

We thank Dr Christos Savva, Dr Emma Hesketh, Dr Claudia Lancey, and Dr TJ Ragan from the Midlands Regional Cryo-EM facility for help with grid preparation, screening, data collections and processing support. A.K.C. thanks the Lister Institute of Preventative Medicine Prize for support of this research. We would also like to thank the Medical Research Council for the standard research grant (MR/X00029X/1). P.F., N.B., J.C., S.B. V.R., J.B.C., and P.C. were supported by the French National Research Agency (ANR-20-CE11-0026). This work was supported by the Fondation ARC (J.C.). P.C. is a scientist from INSERM. We acknowledge the imaging facility TRI, member of the national infrastructure France-BioImaging (https://ror.org/01y7vt929) supported by the French National Research Agency (ANR-24-INBS-0005 FBI BIOGEN). J.B.C. and V.R. thank the I2BC Protex platform supported by French Infrastructure for Integrated Structural Biology (FRISBI) ANR-10-INBS-0005. J.C. and V.R. were supported by ANR-21-CE12-0019-01, ANR-22-CE12-0037, and ANR 23-CE11-0033.

## Author contributions

A.K.C. directed the study. A.K.C., P.C., and P.F. led the experimental design. H.A., S.Z., A.K.C., S.B., P.F., and P.C. prepared the manuscript. H.A. and S.Z. collected the cryo-EM data and modeled and analyzed the structures. C.H. helped with structural analysis and edited manuscript. G.M. Purified proteins and collected initial cryo-EM data. P.F. designed and built the cellular tools. P.F. and N.B. designed, validated, and carried out the in vivo repair experiments. P.F. and J.C. performed the multi-photon laser micro-irradiation experiments. A.K.C., S.B., and P.C. acquired funding. S.W.H. and D.Y.C. advised with cryo-EM data collection set-up and processing. S.W.H. helped with structural analysis and manuscript preparation. V.R. and J.B.C. provided insect cells pellets for human NHEJ factors.

## Competing interests

The authors declare no competing interests.
