## [Transparent Peer Review file · Nature Communications]

Structural and Functional Insights into the Interaction between Ku70/80 and Pol X Family Polymerases in NHEJ

Corresponding Author: Dr Amanda Chaplin

Version 0:

Reviewer comments:

Reviewer #1

(Remarks to the Author)

The authors use an assemblage of NHEJ proteins and carry out cryo-EM. Within that assemblage, they focus on possible interaction of polymerase lambda and Ku. They highlight possible residues of interaction within Ku and polymerase lambda. In cellular studies, they observe sites of localization of Ku and pol lambda and various point mutant versions of these after irradiation or transient transfection of a plasmid substrate into 293 cells. Possible contact points of Ku and pol lambda are mutated and the impact on the number and intensity of foci of accumulation of fluorescently labeled version are assessed.

I have the following points listed that I would like the authors to consider and address.

1. I could not discern the path of the DNA in the authors cryo-EM figures.
2. Purity of the proteins used in the cryo-EM studies must be provided in the form of full-length protein gels to exclude effects of contaminating bands.
3. The authors cryoEM work did not consider omitting DNA-PKcs in their assemblies. This would be very important, since Ku + pol mu alone can form short range or close synaptic complexes on DNA ends with merely one G:C base pair of end-to-end alignment, based on smFRET with purified proteins (PMID 32052035). Therefore, it is important that the authors examine simpler assemblies. Importantly, in the PMID 32052035 studies, DNA-PKcs not only was not supportive for noncovalent synapsis, but it was actually slightly inhibitory for smFRET synapsis in this study using purified proteins. Though analysis of simpler assemblies for this paper may seem like a lot of work now, it may be very fruitful for the authors if they actually follow this suggestion.
4. In PMID, 15574326, Figure1 showed ensemble (bulk solution) interaction of purified pol mu and Ku. In the Supplemental Figure 5 of that same paper, purified pol lambda associates with the Ku:DNA complex. Reactions were done in the same way as in Figure 1 of that paper. The supershifted species in lane 5 of that Suppl. Fig. 5 were pol lambda:Ku:DNA complexes. Adding increasing amounts of anti-pol lambda polyclonal Ab specifically diminished the supershifted bands and caused retention of the 60 bp DNA probe in the wells as shown in lane 6, 7 and 8 of that Suppl. Fig. 5 study. The Ku:DNA complexes were not diminished based on quantitation on a phosphorimager. In light of the results just described with two human pol X polymerases + Ku (without other proteins), this is an additional reason why it is important for cryo-EM studies to be done with these simpler assemblies before jumping to complex assemblies. This is particularly important because Juan He has said in recent seminars that his lab observes a large number of different interaction interfaces between any two DNA-PKcs molecules. This raises the question for everyone in the NHEJ field as to which of these DNA-PKcs dimerization surfaces is specific.
5. An additional reason for considering the simpler assemblies for structural studies is that invertebrates have not been demonstrated to have any DNA-PKcs enzymatic activity to date. Though there are focal amino acid homologies of DNA-PKcs present in possible PI3K-related precursor genes in invertebrates, there has not been any enzymatic activity demonstrated for DNA-PKcs. This means that NHEJ is likely occurring in the absence of DNA-PKcs enzymatic activity in an enormous number of organisms (more than the number of vertebrate organisms). As if these points were not sufficient, yeast carries out NHEJ without DNA-PKcs as well.

Reviewer #2

(Remarks to the Author)

In the submitted manuscript, Frit et al report on the initial interaction required to recruit DNA polymerase lambda to the Non-Homologous End Joining Complex (NHEJ) to repair double strand breaks (DSB). NHEJ is a predominant repair mechanism by which DSBs are repaired which means novel findings in this area are of broad interest. In this study, Frit et al report on a low resolution structures of the long range synaptic complex using cryoEM that reveals the binding location of the BRCT domain of polymerase lambda (whole protein was used, but catalytic domain was not visible) to the Ku70/Ku80 dimer. While cryoEM structures of the long and short range NHEJ complexes have been solved and reported previously, none of the complexes have a polymerase present. Frit et al generate point mutations on both the BRCT domain of polymerase lambda and Ku70/Ku80 to validate their low resolution model using both a protein recruitment assay initiated by biphoton laser-damage and a gap-filling reporter assay. I find this manuscript to be well written and easy to follow and the experiments justify the claims. The results represent an important incremental step in understanding how DNA polymerase lambda is recruited to the long range NHEJ complex to help initiate the gap filling process. This work however does not explain how the catalytic domain of the polymerase interacts with the NHEJ complex to fill the gaps. While what is presented is important and well characterized, in my opinion, I do not feel it rises to the expected impact of a Nat Comm manuscript. Obviously, obtaining a short range complex with the catalytic domain of lambda engaged at the gap would clear the publication bar, but short of that, one thing that could be done to raise the value of the manuscript would be to test the proposed "unified model" of polymerase X proteins interacting with the NHEJ complex. This could be done by obtaining cryoEM structures of Pol Mu or TdT bound to the long range complex (even if the catalytic domains are not ordered/visible) confirming a similar interaction region as proposed by the AlphaFold model presented in this manuscript or by using reporter assays to demonstrate that the mutations in the BRCT domains and Ku70/Ku80 disrupt repair by these other X-family polymerases.

Other improvements

- 1) Please include the Expression and Purification methods for Pol lambda and PAXX in the Materials and Methods section.
- 2) Please include methods for the transfection used in the ionizing radiation and cell survival analysis method.
- 3) Pg9 lines 223 and 230: Should Ku70 T307 also be listed here? It looks like it also greatly diminished recruitment of the BRCT domain of Pol lambda
- 4) Pg15 line 385: Can you show the possible weak interaction region between the BRCT domain of Pol Lambda and DNA-PKcs in a figure? This figure could also be referenced on pg 17 line 426.
- 5) Pg16 line 407: as suggested above, test the "unified model" with pol Mu and Tdt.
- 6) Pg18 figure 5: Its hard to tell from the figure, but it looks like pol Mu and TdT BRCT domains are in different orientations in the AlphaFold model with respect to Lambda's. If they are different, does the orientation of the model make sense with respect to mutations that have been done previously on the BRCT domains of these proteins (DeRose, 2007 in Biochemistry)?
- 7) Pg18 Figure 5D: Please enlarge. Too difficult to read. Also true for figure S7F.
- 8) Pg 19 line 482: Don't forget to add the accession codes here (already in table).
- 9) Error bars in the bar graphs are difficult to see. Please make more visible.

Reviewer #3

(Remarks to the Author)

Frit et al. have studied the structure of the Ku70/80-DNA-PKcs-pol λ complex and identified the specific region within the BRCT domain of pol λ that is responsible for binding to Ku70/80 in the DNA-PK complex. Through mutagenesis studies, they pinpointed key residues that mediate the interaction between the BRCT domain of pol λ and the bridging region of Ku70/80. This research provides valuable insights into the interplay between the end-processing enzyme pol λ and the initial non-homologous end joining (NHEJ) complex, DNA-PK.

I have a few minor suggestions:

In lines 142-144, the authors mention that the catalytic domain of pol λ does not stably interact with DNA. Could the authors speculate on how pol λ recognizes the primer/template structure and mediates fill-in end-processing within the long-range (LR) complex? Without this, the biological relevance of the structure is difficult to grasp. The reason the authors did not observe the interaction between pol λ and DNA might be the use of blunt-end DNA in this study. However, blunt-end DNA is not the physiological substrate of pol λ . I suggest using substrates with microhomologies, as done by Dr. Pedersen (PMID: 35778389), for the structural study.

In Figure S7H, could the authors clarify why gap-filling activity increases in pol μ knockout cells compared to control cells?

Reviewer #4

(Remarks to the Author)

The authors describe an in-depth study of interactions within the NHEJ synaptic complex, including 2 cryoEM structures (resolution 4.2-4.6 Å) and a large number of protein-protein interaction assays. The specificity of binding for specific factors, and the amino acids essential for these interactions, are determined. The manuscript represents a significant contribution to the field, the experiments are exhaustive and well-performed. In all I recommend its publication. However, many of the sentences are too long and complex, and could be simplified to improve comprehension. The authors should also be careful to define all abbreviations when they first appear - for example, PK in the Abstract. I would also suggest asking a native English speaker to read through the text, as there are a significant number of mistakes and awkward phrasings.

Version 1:

Reviewer comments:

Reviewer #1

(Remarks to the Author)

I am satisfied with the revisions.

Reviewer #2

(Remarks to the Author)

The authors have made significant changes to the manuscript including additional CryoEM and biochemical/cellular data supporting a unified model for the interaction of X family polymerases via the BRCT domains with the same site of Ku70/80. They also have addressed all of my other concerns. I now feel this manuscript meets the high standards of Nature Communications and I recommend it for publication.

A couple of editorial notes:

Pg 5 line 124: Maybe should read "...once either polymerase concentration was increased."

Pg 5 line 125: Should Figure S1D also be listed along with S1C?

Pg 10 line 290: Due to the discussion of R43, I think the position of R43 needs to be highlighted somehow in Figure 5B

Pg 11 line 336-337: Residues 128-130 are listed but residues 130 and 133 are what are shown in Figure S11C. Can you make consistent?

Pg 12 line 632: Should this read "Cryo-EM grid preparation of Ku 70/80-DNA and polu" ?

Figure S8: this figure appears to be missing the western-blot for panel C.

Figure S12C: Since you are labeling the AlphaFold models of lambda and mu with AF3, I think in figure S12C TdT should also be labeled as "TdT AF3" to be consistent.

Table S1: The wrong symbol is used for Mu. Since you are using L for lambda, maybe just make the label PoIM ?

Reviewer #3

(Remarks to the Author)

The authors have addressed all of my concerns.

Reviewer #4

(Remarks to the Author)

The authors have worked extensively to revise the manuscript, and have answered all this reviewer's requests. In addition, the writing is considerably improved. It is my opinion that this work represents a major contribution to the field. In its present state, I recommend publication.

Response to Reviewers

We thank all the reviewers for their careful consideration of the manuscript and were pleased to read that they agreed the results represent an important incremental step in understanding how DNA polymerases are recruited to the NHEJ complexes. Thanks to their comments, we have added additional data to the manuscript : i) the cryo-EM structure of Pol Mu only with Ku-DNA as asked by reviewers 1&2 (new Fig. 5 A-C and Figs. S1, S9-12), ii) in cellulose gap-filling data with Ku80 mutants at the Ku-PolX interface that complete the data with Ku70 mutants (new Fig. 4F and Fig. S8B), and iii) gap-filling data with Pol mu mutants as deduced from our new structure and as asked by reviewer 2 (new Fig. 5D and Fig. S8C). Our responses to the reviewer's comments are detailed below.

Reviewer #1 (Remarks to the Author)

The authors use an assemblage of NHEJ proteins and carry out cryo-EM. Within that assemblage, they focus on possible interaction of polymerase lambda and Ku. They highlight possible residues of interaction within Ku and polymerase lambda. In cellular studies, they observe sites of localization of Ku and pol lambda and various point mutant versions of these after irradiation or transient transfection of a plasmid substrate into 293 cells. Possible contact points of Ku and pol lambda are mutated and the impact on the number and intensity of foci of accumulation of fluorescently labeled version are assessed.

I have the following points listed that I would like the authors to consider and address.

1. I could not discern the path of the DNA in the authors cryo-EM figures.

We thank the reviewer for this comment and agree the DNA is difficult to see in such a large multi-protein complexes. We have now added additional DNA labels and arrows to indicate where the DNA is within the complexes and hope this is now clearer to the reader of the manuscript. If still difficult we can easily change the colour or orientation of the images displayed.

2. Purity of the proteins used in the cryo-EM studies must be provided in the form of full-length protein gels to exclude effects of contaminating bands.

This is indeed a necessity with *in vitro* structural studies and have now provided an SDS-PAGE gel image in Figure S1B showing the purity of the complex collected by cryo-EM.

3. The authors cryoEM work did not consider omitting DNA-PKcs in their assemblies. This would be very important, since Ku + pol mu alone can form short range or close synaptic complexes on DNA ends with merely one G:C base pair of end-to-end alignment, based on smFRET with purified proteins (PMID 32052035). Therefore, it is important that the authors examine simpler assemblies. Importantly, in the PMID 32052035 studies, DNA-PKcs not only was not supportive for noncovalent synapsis, but it was actually slightly inhibitory for smFRET synapsis in this study using purified proteins.

Though analysis of simpler assemblies for this paper may seem like a lot of work now, it may be very fruitful for the authors if they actually follow this suggestion.

We agree with the reviewer's comment here and have now included additional data of only Ku with microhomology DNA (from the reference mentioned above, PMID: 32052035, now cited in the manuscript) with Polymerase Mu (Figure 5 and S9-11).

We thank the reviewer because before trying this DNA, we could not observe extra density for Pol Mu or Pol λ with only Ku70/80. We are pleased that our study now shows the interaction of the polymerases within the long-range synaptic complex and within simpler assemblies of only Ku70/80 and polymerases.

Future studies investigating the role of DNA-Polymerases within the short-range complex will indeed be very interesting, however this involves a huge amount of work and for this present study we wanted to focus more on the specific interaction between the BRCT domain of polymerases and Ku70/80. We feel that although further studies will be fruitful, it is important for the DNA-repair field to publish this important interaction interface.

4. In PMID, 15574326, Figure 1 showed ensemble (bulk solution) interaction of purified pol mu and Ku. In the Supplemental Figure 5 of that same paper, purified pol lambda associates with the Ku:DNA complex. Reactions were done in the same way as in Figure 1 of that paper. The supershifted species in lane 5 of that Suppl. Fig. 5 were pol lambda:Ku:DNA complexes. Adding increasing amounts of anti-pol lambda polyclonal Ab specifically diminished the supershifted bands and caused retention of the 60 bp DNA probe in the wells as shown in lane 6, 7 and 8 of that Suppl. Fig. 5 study. The Ku:DNA complexes were not diminished based on quantitation on a phosphorimager.

In light of the results just described with two human pol X polymerases + Ku (without other proteins), this is an additional reason why it is important for cryo-EM studies to be done with these simpler assemblies before jumping to complex assemblies. This is particularly important because Juan He has said in recent seminars that his lab observes a large number of different interaction interfaces between any two DNA-PKcs molecules. This raises the question for everyone in the NHEJ field as to which of these DNA-PKcs dimerization surfaces is specific.

We thank the reviewer for this suggestion and have now added in additional data of Ku with Pol Mu (as mentioned above). We now also include EMSA gels showing the interaction between Ku70/80 and Pol λ and μ (**Figure S1C, D**).

It is true that DNA-PKcs alone can dimerize, however when in the context of DNA-PK, limited dimeric assemblies have been identified, two of which we have shown have biological importance in DNA repair^{1,2}. We believe the Ku80-mediated and XLF-mediated DNA-PK dimers are important for the repair process, and in addition to Ku70/80 being able to act as a “hub” for NHEJ factors, we are more recently observing that DNA-PK can indeed also fulfil this function (unpublished). Therefore, although the polymerases may not act on the DNA ends within the long-range complexes, their presence in these complexes, even if a movement is required when DNA-PKcs binds, illustrates the importance of their initial anchoring.

5. An additional reason for considering the simpler assemblies for structural studies is that invertebrates have not been demonstrated to have any DNA-PKcs enzymatic activity to date. Though there are focal amino acid homologies of DNA-PKcs present in possible PI3K-related precursor genes in invertebrates, there has not been any enzymatic activity demonstrated for DNA-PKcs. This means that NHEJ is likely occurring in the absence of DNA-PKcs enzymatic activity in an enormous number of organisms (more than the number of vertebrate organisms). As if these points were not sufficient, yeast carries out NHEJ without DNA-PKcs as well.

We agree many other systems are able to carry out the NHEJ mechanism without DNA-PKcs and we have now included further data of Ku70/80 with Pol μ . We did try to determine the structure of Ku70/80 with Pol λ , however, we were unable to see additional density in this simplified system, even with different DNA types tested.

Reviewer #2 (Remarks to the Author)

In the submitted manuscript, Frit et al report on the initial interaction required to recruit DNA polymerase lambda to the Non-Homologous End Joining Complex (NHEJ) to repair double strand breaks (DSB). NHEJ is a predominant repair mechanism by which DSBs are repaired which means novel findings in this area are of broad interest. In this study, Frit et al report on a low resolution structures of the long range synaptic complex using cryoEM that reveals the binding location of the BRCT domain of polymerase lambda (whole protein was used, but catalytic domain was not visible) to the Ku70/Ku80 dimer. While cryoEM structures of the long and short range NHEJ complexes have been solved and reported previously, none of the complexes have a polymerase present. Frit et al generate point mutations on both the BRCT domain of polymerase lambda and Ku70/Ku80 to validate their low resolution model using both a protein recruitment assay initiated by biphoton laser-damage and a gap-filling reporter assay. I find this manuscript to be well written and easy to follow and the experiments justify the claims. The results represent an important incremental step in understanding how DNA polymerase lambda is recruited to the long range NHEJ complex to help initiate the gap filling process. This work however does not explain how the catalytic domain of the polymerase interacts with the NHEJ complex to fill the gaps. While what is presented is important and well characterized, in my opinion, I do not feel it rises to the expected impact of a Nat Comm manuscript. Obviously, obtaining a short range complex with the catalytic domain of lambda engaged at the gap would clear the publication bar, but short of that, one thing that could be done to raise the value of the manuscript would be to test the proposed “unified model” of polymerase X proteins interacting with the NHEJ complex. This could be done by obtaining cryoEM structures of Pol Mu or TdT bound to the long range complex (even if the catalytic domains are not ordered/visible) confirming a similar interaction region as proposed by the Alphafold model presented in this manuscript or by using reporter assays to demonstrate that the mutations in the BRCT domains and Ku70/Ku80 disrupt repair by these other X-family polymerases.

We thank the reviewer for their careful consideration of the manuscript and literature of the DNA repair field. We again thank the reviewer for their comment that this is an important incremental step in understanding how DNA polymerase lambda is recruited to the long range NHEJ complex to help initiate the gap filling process. We agree this work does not describe how the catalytic domain of the polymerase interacts with the NHEJ complex, however in this present study the focus was to determine how the polymerase initially binds and interacts with the NHEJ machinery, which we have determined. We have now included data of Ku with Pol Mu, and discussed differences in the orientation of the BRCT domain on Ku70/80 alone and when in complex with DNA-PK. We agree the short-range complex would indeed be very important to determine but is an additional challenging piece of work which ourselves and others are undertaking. We must also note here that we did try cryo-EM structural studies with TdT and Pol Mu within the long-range complexes however we did not observe any additional density where we identified the polymerase lambda density. Further

optimisation with different conditions, DNA substrate and/or protein partners are obviously required in the future.

According to the reviewer suggestions, we have used our gap-filling reporter assays in double Pol L + Pol M KO cells transfected with Pol mu and have assessed the impact of mutations in the BRCT domain of Pol mu on the Pol mu-mediated gap-filling activity, based on our new structure of Pol mu interacting with Ku and on sequences comparison, (new **Figure 5D**). The data clearly validate the key positions at the pol mu-Ku interface. We believe that altogether our structural and cellular data support the unified model of Pol X proteins interacting with the NHEJ complex that we propose (**Figure S12**) and have changed the manuscript title accordingly.

Other improvements

1) Please include the Expression and Purification methods for Pol lambda and PAXX in the Materials and Methods section.

We apologise to the reviewer for missing this detail and have now included the materials and methods for these protein purifications.

2) Please include methods for the transfection used in the ionizing radiation and cell survival analysis method.

We apologise if the description of the cells used for the cell survival experiments was unclear. Actually, all cell lines used in the study were stably modified following lentiviral transduction, with the exception of KO cells that were obtained by transient transfection with vectors expressing eCas9/gRNA. This is described in the Materials and Methods section and the corresponding cited references.

More specifically, the U2OS cells expressing Ku70 mutants used in the cell survival assay in the new Fig 5E were those also used for the laser micro-irradiation assay (**Fig. 3B**); similarly, the HEK-293T cells expressing the Pol lambda mutants used in the cell survival assay in the new Fig 5G were also those used for the gap-filling assay (see new **Fig. 5F** and **Figs. 3C, D**). These points have also been added to the corresponding figure legends.

3) Pg9 lines 223 and 230: Should Ku70 T307 also be listed here? It looks like it also greatly diminished recruitment of the BRCT domain of Pol lambda

We apologise for this omission and have now added this position in the corresponding sentences.

4) Pg15 line 385: Can you show the possible weak interaction region between the BRCT domain of Pol Lambda and DNA-PKcs in a figure? This figure could also be referenced on pg 17 line 426.

This is important, and we have now included a supplemental figure (Figure S11C) showing the potential interaction residues and discussed this interaction in the discussion section.

5) Pg16 line 407: as suggested above, test the “unified model” with pol Mu and Tdt.

As mentioned above we have now included a new structure of Ku70/80 with Pol Mu and discussed the differences and similarities to Pol λ and TdT.

6) Pg18 figure 5: Its hard to tell from the figure, but it looks like pol Mu and TdT BRCT

domains are in different orientations in the AlphaFold model with respect to Lambda's. If they are different, does the orientation of the model make sense with respect to mutations that have been done previously on the BRCT domains of these proteins (DeRose, 2007 in Biochemistry)?

We have now included additional AlphaFold comparisons with our existing and new experimental data (Figure S12). We also now discuss the sequence alignment in Figure S11, describing the conservation of the residues across the Pol X family members.

7) Pg18 Figure 5D: Please enlarge. Too difficult to read. Also true for figure S7F. These figures have now been enlarged (Figures S11B and S7F).

8) Pg 19 line 482: Don't forget to add the accession codes here (already in table). Thank you for noticing, these have now been included here, with the addition of the new data too.

9) Error bars in the bar graphs are difficult to see. Please make more visible. We have corrected the graphs accordingly.

Reviewer #3 (Remarks to the Author):

Frit et al. have studied the structure of the Ku70/80-DNA-PKcs-pol λ complex and identified the specific region within the BRCT domain of pol λ that is responsible for binding to Ku70/80 in the DNA-PK complex. Through mutagenesis studies, they pinpointed key residues that mediate the interaction between the BRCT domain of pol λ and the bridging region of Ku70/80. This research provides valuable insights into the interplay between the end-processing enzyme pol λ and the initial non-homologous end joining (NHEJ) complex, DNA-PK.

We thank the reviewer for the positive comment that this research provides valuable insights into end-processing enzymes in NHEJ.

I have a few minor suggestions:

In lines 142-144, the authors mention that the catalytic domain of pol λ does not stably interact with DNA. Could the authors speculate on how pol λ recognizes the primer/template structure and mediates fill-in end-processing within the long-range (LR) complex? Without this, the biological relevance of the structure is difficult to grasp. The reason the authors did not observe the interaction between pol λ and DNA might be the use of blunt-end DNA in this study. However, blunt-end DNA is not the physiological substrate of pol λ . I suggest using substrates with microhomologies, as done by Dr. Pedersen (PMID: 35778389), for the structural study.

We believe that this complex is an initial recognition site of polymerase λ and μ . That the BRCT interaction with DNA-PK enables polymerase λ to be primed to enable the catalytic domain to carry out its function, following DNA-PKcs activation and likely removal with ATP. We have now added some additional text within the manuscript to explain this. We also believe at this step of the mechanism the DNA end is blocked by the central helix of DNA-PKcs and once the mechanism progresses the polymerase would be able to recognise the

ends. We have now included new cryo-EM data of Ku70/80 with microhomology DNA (as the reviewer mentioned) with polymerase μ (**Figure 5**). We thank the reviewer for this suggestion and agree that DNA end configuration is very important for the polymerases. We should note that we tried Pol λ with Ku70/80 with different DNA substrates and could only observe the BRCT within the long-range complex. Further optimisation with different conditions, DNA substrate and/or protein partners are obviously required in the future.

In Figure S7H, could the authors clarify why gap-filling activity increases in pol μ knockout cells compared to control cells?

The substrate that we designed is more suited for Pol lambda gap-filling activity than for Pol mu (PMID:26240371; PMID: 31862156). Indeed, we found that in a POL L KO background, Pol mu only marginally fills the gap on this substrate (Figure S7H) and even under condition of Pol mu over-expression (Figure S8C), Pol mu fills it not optimally (Figure 5D). Given that both polymerases occupy the same site on the Ku bridge (Figure 5C), we believe that in WT cells, the Pol mu activity that is unsuited on this substrate exerts a partial negative effect on Pol lambda-mediated gap-filling activity, so that the knock-out of Pol mu releases this effect and boosts gap-filling by Pol lambda. Similarly, overexpression of Pol lambda also boosts gap-filling on this substrate (Figure S7H), likely through competition with Pol mu binding to Ku.

Reviewer #4 (Remarks to the Author):

The authors describe an in-depth study of interactions within the NHEJ synaptic complex, including 2 cryoEM structures (resolution 4.2-4.6 Å) and a large number of protein-protein interaction assays. The specificity of binding for specific factors, and the amino acids essential for these interactions, are determined. The manuscript represents a significant contribution to the field, the experiments are exhaustive and well-performed. In all I recommend its publication. However, many of the sentences are too long and complex, and could be simplified to improve comprehension. The authors should also be careful to define all abbreviations when they first appear - for example, PK in the Abstract. I would also suggest asking a native English speaker to read through the text, as there are a significant number of mistakes and awkward phrasings.

We thank the reviewer for stating that the manuscript represents a significant contribution to the field and experiments are exhaustive and well performed. We have now gone through the manuscript again, and as one of the corresponding authors is a native English speaker, have corrected any long sentences or awkward phrasings. We have also checked any abbreviations.

Useful References

1. Chaplin, A.K. et al. Dimers of DNA-PK create a stage for DNA double-strand break repair. *Nat Struct Mol Biol* **28**, 13-19 (2020).
2. Chaplin, A.K. et al. Cryo-EM of NHEJ supercomplexes provides insights into DNA repair. *Mol Cell* **81**, 3400-3409 e3 (2021).

REVIEWERS' COMMENTS

Reviewer #1 (Remarks to the Author):

I am satisfied with the revisions.

Reviewer #2 (Remarks to the Author):

The authors have made significant changes to the manuscript including additional CryoEM and biochemical/cellular data supporting a unified model for the interaction of X family polymerases via the BRCT domains with the same site of Ku70/80. They also have addressed all of my other concerns. I now feel this manuscript meets the high standards of Nature Communications and I recommend it for publication.

A couple of editorial notes:

Pg 5 line 124: Maybe should read "...once either polymerase concentration was increased." **This has now been changed.**

Pg 5 line 125: Should Figure S1D also be listed along with S1C? **Added.**

Pg 10 line 290: Due to the discussion of R43, I think the position of R43 needs to be highlighted somehow in Figure 5B. **We have now added this residue as a stick and labelled.**

Pg 11 line 336-337: Residues 128-130 are listed but residues 130 and 133 are what are shown in Figure S11C. Can you make consistent? **Now made consistent.**

Pg 12 line 632: Should this read "Cryo-EM grid preparation of Ku 70/80-DNA and polu" ? **Now changed.**

Figure S8: this figure appears to be missing the western-blot for panel C. **This has been added.**

Figure S12C: Since you are labeling the AlphaFold models of lambda and mu with AF3, I think in

figure S12C TdT should also be labeled as "TdT AF3" to be consistent. **Now labelled.**

Table S1: The wrong symbol is used for Mu. Since you are using L for lamda, maybe just make the label PolM ? **Now made consistent.**

Reviewer #3 (Remarks to the Author):

The authors have addressed all of my concerns.

Reviewer #4 (Remarks to the Author):

The authors have worked extensively to revise the manuscript, and have answered all this reviewer's requests. In addition, the writing is considerably improved. It is my opinion

that this work represents a major contribution to the field. In its present state, I recommend publication.